# Generative Semi-supervised Graph Anomaly Detection

**Hezhe Qiao[1], Qingsong Wen[2], Xiaoli Li[3,4], Ee-Peng Lim[1], Guansong Pang[1]***

[1]School of Computing and Information Systems, Singapore Management University
[2]Squirrel AI
[3] Institute for Infocomm Research, A*STAR, Singapore
[4] A*STAR Centre for Frontier AI Research, Singapore
`hezheqiao.2022@phdcs.smu.edu.sg, qingsongedu@gmail.com`
`xlli@i2r.a-star.edu.sg, eplim@smu.edu.sg, gspang@smu.edu.sg`

## Abstract

This work considers a practical semi-supervised graph anomaly detection (GAD) scenario, where part of the nodes in a graph are known to be normal, contrasting to the extensively explored unsupervised setting with a fully unlabeled graph. We reveal that having access to the normal nodes, even just a small percentage of normal nodes, helps enhance the detection performance of existing unsupervised GAD methods when they are adapted to the semi-supervised setting. However, their utilization of these normal nodes is limited. In this paper we propose a novel Generative GAD approach (namely **GGAD**) for the semi-supervised scenario to better exploit the normal nodes. The key idea is to generate pseudo anomaly nodes, referred to as *outlier nodes*, for providing effective negative node samples in training a discriminative one-class classifier. The main challenge here lies in the lack of ground truth information about real anomaly nodes. To address this challenge, GGAD is designed to leverage two important priors about the anomaly nodes – *asymmetric local affinity* and *egocentric closeness* – to generate reliable outlier nodes that assimilate anomaly nodes in both graph structure and feature representations. Comprehensive experiments on six real-world GAD datasets are performed to establish a benchmark for semi-supervised GAD and show that GGAD substantially outperforms state-of-the-art unsupervised and semi-supervised GAD methods with varying numbers of training normal nodes. Code is available at https://github.com/mala-lab/GGAD.

## 1 Introduction

Graph anomaly detection (GAD) has received significant attention due to its broad application domains, *e.g.*, cyber security and fraud detection [9, 16, 33]. However, it is challenging to recognize anomaly nodes in a graph due to its complex graph structure and attributes [27, 34, 44, 46, 62, 65, 66]. Moreover, most traditional anomaly detection methods [4, 39, 60] are designed for Euclidean data, which are shown to be ineffective on non-Euclidean data like graph data [8, 17, 28, 44, 51, 66]. To address this challenge, as an effective way of modeling graph data, graph neural networks (GNN) have been widely used for deep GAD [34, 45]. These GNN methods typically assume that the labels of all nodes are unknown and perform anomaly detection in a fully unsupervised way by, *e.g.*, data reconstruction [8, 11], self-supervised learning [6, 28, 36, 61], or one-class homophily modeling [44]. Although these methods achieve remarkable advances, they are not favored in real-world applications where the labels for normal nodes are easy to obtain due to their overwhelming presence in a graph. This is because their capability to utilize those labeled normal nodes is very limited due to their

---

*Corresponding author: G. Pang

38th Conference on Neural Information Processing Systems (NeurIPS 2024).

inherent unsupervised nature. There have been some GAD methods [12, 16, 26, 34, 49, 50, 52, 54] designed in a semi-supervised setting, but their training relies on the availability of both labeled normal and anomaly nodes, which requires a costly annotation of a large set of anomaly nodes. This largely restricts the practical application of these methods.

Different from the aforementioned two GAD settings, this paper instead considers a practical yet under-explored semi-supervised GAD scenario, where part of the nodes in the graph are known to be normal. Such a one-class classification setting has been widely explored in anomaly detection on other data types, such as visual data [4, 39], time series [57], and tabular data [19], but it is rarely explored in anomaly detection on graph data. Recently there have been a few relevant studies in this line [3, 27, 31, 38, 56], but they are on graph-level anomaly detection, *i.e.*, detecting abnormal graphs from a set of graphs, while we explore the semi-supervised setting for abnormal node detection. We establish an evaluation benchmark for this problem and show that having access to these normal nodes helps enhance the detection performance of existing unsupervised GAD methods when they are properly adapted to the semi-supervised setting (see Table 1). However, due to their original unsupervised designs, they cannot make full use of these labeled normal nodes.

To better exploit those normal nodes, we propose a novel generative GAD approach, namely **GGAD**, aiming at generating pseudo anomaly nodes, referred to as **outlier nodes**, for providing effective negative node samples in training a discriminative one-class classifier on the given normal nodes. The key challenge in this type of generative approach is the absence of ground-truth information about real anomaly nodes. There have been many generative anomaly detection approaches that learn adversarial outliers to provide some weak supervision of abnormality [37, 47, 59, 64], but they are designed for non-graph data and fail to take account of the graph structure information in the outlier generation. Some recent methods, such as AEGIS [7], attempt to adapt this approach for GAD, but the outliers are generated based on simply adding Gaussian noises to the GNN-based node representations, ignoring the structural relations between the outliers and the graph nodes. Consequently, the distribution of the generated outliers is often mismatched to that of the real anomaly nodes, as illustrated in Fig. 1a, and demonstrates very different local structure (Fig. 1c).

Our approach GGAD tackles this issue with a method to generate outlier nodes that assimilate

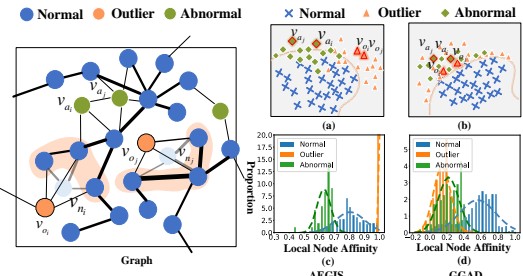

Figure 1: **Left:** An exemplar graph with the edge width indicates the level of affinity connecting two nodes, in which normal nodes (*e.g.*, $v_{n_i}$ and $v_{n_j}$) have stronger affinity to its neighboring normal nodes than anomaly nodes (*e.g.*, $v_{a_i}$ and $v_{a_j}$) due to homophily relation within the normal class. Our approach GGAD aims to generate outliers (*e.g.*, $v_{o_i}$ and $v_{o_j}$) that can well assimilate the anomaly nodes. **Right:** The outliers generated by methods like AEGIS [7] that ignore their structural relation often mismatch the distribution of abnormal nodes (**a**), due to their false local affinity (**c**). By contrast, GGAD incorporates two important priors about anomaly nodes to generate outliers so that they well assimilate the (**b**) feature representation and (**d**) local structure of abnormal nodes.

the anomaly nodes in both local structure and feature representation. It is motivated by two important priors about the anomaly nodes. The first one is an **asymmetric local affinity** phenomenon revealed in recent studies [12, 13, 44], *i.e.*, *the affinity between normal nodes is typically significantly stronger than that between normal and abnormal nodes*. Inspired by this, GGAD generates outlier nodes in a way to enforce that they have a smaller local affinity to their local neighbors than the normal nodes. This objective aligns the distribution of the outlier nodes to that of the anomaly nodes in terms of graph structure. The second prior knowledge is that many anomaly nodes exhibit high similarity to the normal nodes in the feature space due to its subtle abnormality [24, 44] or adversarial camouflage [10, 14, 29, 49]. We encapsulate this prior knowledge as **egocentric closeness**, mandating that *the feature representation of the outlier nodes should be closed to the normal nodes that share similar local structure as the outlier nodes*. GGAD incorporates these two priors through two loss functions to generate outlier nodes that are well aligned to the distribution of the anomaly nodes in both local structure affinity (see Fig. 1d) and feature representation (see Fig. 1b). We can then train a

discriminative one-class classifier on the labeled normal nodes, with these generated outlier nodes treated as the negative samples.

Accordingly, our main contributions can be summarized as follows:

- We explore a practical yet under-explored semi-supervised GAD problem where part of the normal nodes are known, and establish an evaluation benchmark for the problem.
- We propose a novel generative GAD approach, GGAD, for the studied setting. To the best of our knowledge, it is the first work aiming for generating outlier nodes that are of similar local structure and node representations to the real anomaly nodes. The outlier nodes serve as effective negative samples for training a discriminative one-class classifier.
- We encapsulate two important priors about anomaly nodes – asymmetric local affinity and egocentric closeness – and leverage them to introduce an innovative outlier node generation method. Although these priors may not be exhaustive, they provide principled guidelines for generating learnable outlier nodes that can well assimilate the real anomaly nodes in both graph structure and feature representation across diverse real-world GAD datasets.
- Extensive experiments on six large GAD datasets demonstrate that our approach GGAD substantially outperforms 12 state-of-the-art unsupervised and semi-supervised GAD methods with varying numbers of training normal nodes, achieving over 15% increase in AU-ROC/AUPRC compared to the best contenders on the challenging datasets.

## 2 Related Work

### 2.1 Graph Anomaly Detection

Numerous graph anomaly detection methods, including shallow and deep approaches, have been proposed. Shallow methods like Radar [22], AMEN [43], and ANOMALOUS [42] are often bottlenecked due to the lack of representation power to capture the complex semantics of graphs. With the development of GNN in node representation learning, many deep GAD methods show better performance than shallow approaches. Here we focus on the discussion of the deep GAD methods in two relevant settings: unsupervised and semi-supervised GAD.

**Unsupervised Approach.** Existing unsupervised GAD methods are typically built using a conventional anomaly detection objective, such as data reconstruction. The basic idea is to capture the normal activity patterns and detect anomalies that behave significantly differently. As one of the most popular methods, reconstruction-based methods using graph auto-encoder (GAE) have been widely applied for GAD [1]. DOMINANT is the first work that applies GAE on the graph to reconstruct the attribute and structure leveraging GNNs [8]. Fan *et al.* propose AnomalyDAE to further improve the performance by enhancing the importance of the reconstruction on the graph structure. In addition to reconstruction, some methods focus on exploring the relationship in the graph, *e.g.*, the relation between nodes and subgraphs, to train GAD models. Among these methods, Qiao *et al.* propose TAM [44], which maximizes the local node affinity on truncated graphs, achieving good performance on the synthetic dataset and datasets with real anomalies. Although the aforementioned unsupervised methods achieve good performance and help us identify anomalies without any access to class labels, they cannot effectively leverage the labeled nodes when such information is available.

**Semi-Supervised Approach.** The one-class classification under semi-supervised setting has been widely explored in anomaly detection on visual data, but rarely done on the graph data, except [3, 27, 31, 38, 56] that recently explored this setting for graph-level anomaly detection. To detect abnormal graphs, these methods address a very different problem from ours, which is focused on capturing the normality of a set of given normal graphs at the graph level. By contrast, we focus on modeling the normality at the node level. Some semi-supervised methods have been recently proposed for node-level anomaly detection, but they assume the availability of the labels of both normal and anomaly nodes [12, 16, 26, 41, 48, 52]. By contrast, our setting eases this requirement and requires the labeled normal nodes only.

### 2.2 Generative Anomaly Detection

Generative adversarial networks (GANs) provide an effective solution to generate synthetic samples that capture the normal/abnormal patterns [2, 15, 63]. One type of these methods aims to learn latent

features that can capture the normality of a generative network [30,58]. Methods like ALAD [59], Fence GAN [37] and OCAN [64] are early methods in this line, aiming at making the generated samples lie at the boundary of normal data for more accurate anomaly detection. Motivated by these methods, a similar approach has also been explored in graph data, like AEGIS [7] and GAAN [6] which aim to simulate some abnormal features in the representation space using GNN, but they are focused on adding Gaussian noise to the representations of normal nodes without considering graph structure information. They are often able to generate pseudo anomaly node representations that are separable from the normal nodes for training their detection model, but the pseudo anomaly nodes are mismatched with the distribution of the real anomaly nodes.

## 3 Methodology

### 3.1 Problem Statement

**Semi-supervised GAD**. We focus on the semi-supervised anomaly detection on the attributed graph given some labeled normal nodes. An attributed graph can be denoted by $\mathcal{G} = (\mathcal{V}, \mathcal{E}, \boldsymbol{X})$, where $\mathcal{V} = \{v_1, \cdots, v_N\}$ denotes the node set, $\mathcal{E} \subseteq \mathcal{V} \times \mathcal{V}$ with $e \in \mathcal{E}$ is the edge set in the graph. $e_{ij} = 1$ represents there is a connection between node $v_i$ and $v_j$, and $e_{ij} = 0$ otherwise. The node attributes are denoted as $\mathbf{X} \in \mathbb{R}^{N \times F}$ and $\mathbf{A} \in \{0,1\}^{N \times N}$ is the adjacency matrix of $\mathcal{G}$. $\boldsymbol{x}_i \in \mathbb{R}^F$ is the attribute vector of $v_i$ and $\mathbf{A}_{ij} = 1$ if and only if $(v_i, v_j) \in \mathcal{E}$. $\mathcal{V}_n$ and $\mathcal{V}_a$ represent the normal node set and abnormal node set, respectively. Typically the number of normal nodes is significantly greater than the abnormal nodes, *i.e.*, $|\mathcal{V}_n| \gg |\mathcal{V}_a|$. The goal of semi-supervised GAD is to learn an anomaly scoring function $f : \mathcal{G} \to \mathbb{R}$, such that $f(v) < f(v'), \forall v \in \mathcal{V}_n, v' \in \mathcal{V}_a$ given a set of labeled normal nodes $\mathcal{V}_l \subset \mathcal{V}_n$ and no access to labels of anomaly nodes. All other unlabeled nodes, denoted by $\mathcal{V}_u = \mathcal{V} \setminus \mathcal{V}_l$, comprise the test data set.

**Outlier Node Generation**. Outlier generation aims to generate outlier nodes that deviate from the normal nodes and/or assimilate the anomaly nodes. Such nodes can be generated in either the raw feature space or the embedding feature space. This work is focused on the latter case, as it offers a more flexible way to represent relations between nodes. Our goal is to generate a set of outlier nodes from $\mathcal{G}$, denoted by $\mathcal{V}_o$, in the feature representation space, so that the outlier nodes are well aligned to the anomaly nodes, given no access to the ground-truth anomaly nodes.

**Graph Neural Network for Node Representation Learning.** GNN has been widely used to generate the node representations due to its powerful representation ability in capturing the rich graph attribute and structure information. The projection of node representation using a GNN layer can be generally formalized as

$$\mathbf{H}^{(\ell)} = \text{GNN}\left(\mathbf{A}, \mathbf{H}^{(\ell-1)}; \mathbf{W}^{(\ell)}\right), \tag{1}$$

where $\mathbf{H}^{(\ell)} \in \mathbb{R}^{N \times h^{(l)}}, \mathbf{H}^{(\ell-1)} \in \mathbb{R}^{N \times h^{(l-1)}}$ are the representations of all $N$ nodes in the $(\ell)$-th layer and $(\ell-1)$-th layer, respectively, $h^{(l)}$ is the dimensionality size, $\mathbf{W}^{(\ell)}$ are the learnable parameters, and $\mathbf{H}^{(0)}$ is set to $\mathbf{X}$. $\mathbf{H}^{(\ell)} = \{\mathbf{h}_1, \mathbf{h}_2, \dots, \mathbf{h}_N\}$ is a set of representations of $N$ nodes in the last GNN layer, with $\mathbf{h} \in \mathbb{R}^d$. In this paper, we adopt a 2-layer GCN to model the graph.

### 3.2 Overview of the Proposed GGAD Approach

The key insight of GGAD is to generate learnable outlier nodes in the feature representation space that assimilate anomaly nodes in terms of both local structure affinity and feature representation. To this end, we introduce two new loss functions that incorporate two important priors about anomaly nodes – asymmetric local affinity and egocentric closeness – to optimize the outlier nodes. As shown in Fig. 2a, the outlier nodes are first initialized based on the representations of the neighbors of the labeled normal nodes, followed by the use of the two priors on the anomaly nodes. GGAD implements the asymmetric local affinity prior in Fig. 2b that enforces a larger local affinity of the normal nodes than that of the anomaly nodes. GGAD then models the egocentric closeness in Fig. 2c that pulls the feature representations of the outlier nodes to the normal nodes that share the same ego network. These two priors are implemented through two complementary loss functions in GGAD. Minimizing these loss functions optimizes the outlier nodes to meet both anomaly priors. The resulting outlier nodes are lastly treated as negative samples to train a discriminative one-class classifier on the labeled normal nodes, as shown in Fig. 2d. Below we introduce GGAD in detail.

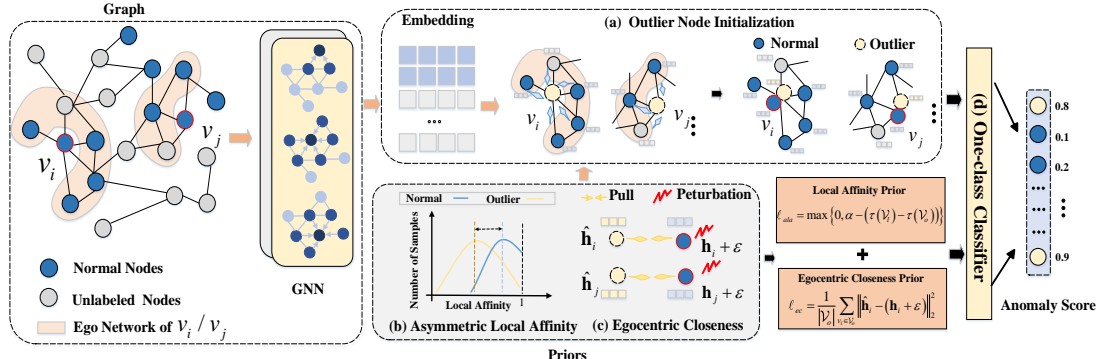

Figure 2: Overview of GGAD. (**a**) It first initializes the outlier nodes based on the feature representations of the ego network of a labeled normal node. We then incorporate the two anomaly node priors (**b-c**) to optimize the outlier nodes so that they are well aligned to the anomalies. (**d**) The resulting generated outlier nodes are treated as negative samples to train a discriminative one-class classifier.

### 3.3  Incorporating the Asymmetric Local Affinity Prior

**Outlier Node Initialization.** Recall that GGAD is focused on generating learnable outlier nodes in the representation space. To enable the subsequent learning of the outlier nodes, we need to produce good representation initialization of the outlier nodes. To this end, we use a neighborhood-aware outlier initialization module that generates the initial outlier nodes' representation based on the representations of the local neighbors of normal nodes. The representations from these neighbor nodes provide an important reference for being normal in a local graph structure. This helps ground the generation of outlier nodes to a real graph structure. More specifically, as shown in Fig. 2a, given a labeled normal node $v_i \in \mathcal{V}_l$ and its ego network $\mathcal{N}(v_i)$ that contains all nodes directly connected with $v_i$, we initialize an outlier node in the representation space by:

$$\hat{\mathbf{h}}_i = \Psi\left(v_i, \mathcal{N}(v_i); \Theta_g\right) = \frac{1}{|\mathcal{N}(v_i)|} \sum_{v_j \in \mathcal{N}(v_i)} \sigma(\tilde{\mathbf{W}}\mathbf{h}_j), \tag{2}$$

where $\Psi$ is a mapping function determined by parameters $\Theta_g$ that contain the learnable parameters $\tilde{\mathbf{W}} \in \mathbb{R}^{d \times d}$ in this module in addition to the parameters $\mathbf{W}^{(\ell)}$ in Eq. (1), and $\sigma(\cdot)$ is an activation function. It is not required to perform Eq. (2) for all training normal nodes. We sample a set of $S$ normal nodes from $\mathcal{V}_l$ and respectively generate an outlier node for each of them based on its ego network. $\hat{\mathbf{h}}_i$ in Eq. (2) serves as an initial representation of the outlier node, upon which two optimization constraints based on our anomaly node priors are devised to optimize the representations of the outlier nodes, as elaborated in the following.

**Enforcing the Structural Affinity Prior.** To incorporate the graph structure prior of anomaly nodes into our outlier node generation, GGAD introduces a local affinity-based loss to enforce the fact that the affinity of the outlier nodes to their local neighbors should be smaller than that of the normal nodes. More specifically, the local node affinity of $v_i$, denoted as $\tau(v_i)$, is defined as the similarity to its neighboring nodes:

$$\tau(v_i) = \frac{1}{|\mathcal{N}(v_i)|} \sum_{v_j \in \mathcal{N}(v_i)} \text{sim}(\mathbf{h}_i, \mathbf{h}_j), \tag{3}$$

The asymmetric local affinity loss is then defined by a margin loss function based on the affinity of the normal nodes and the generated outlier nodes as follows:

$$\ell_{ala} = \max\left\{0, \alpha - \left(\tau(\mathcal{V}_l) - \tau(\mathcal{V}_o)\right)\right\}, \tag{4}$$

where $\tau(\mathcal{V}_o) = \frac{1}{|\mathcal{V}_o|} \sum_{v_i \in \mathcal{V}_o} \tau(v_i)$ and $\tau(\mathcal{V}_l) = \frac{1}{|\mathcal{V}_l|} \sum_{v_i \in \mathcal{V}_l} \tau(v_i)$ represent the average local affinity of the outliers and normal nodes respectively, and $\alpha > 0$ is a hyperparameter controlling the margin between the affinities of these two types of nodes. Eq. (4) enforces this prior at the node set level rather than at each individual outlier node, as the latter case would be highly computationally costly when $\mathcal{V}_l$ or $\mathcal{V}_o$ is large.

## 3.4 Incorporating the Egocentric Closeness Prior

The outliers generated by solely using this local affinity prior may distribute far away from the normal nodes in the representation space, as shown in Fig. 3a. For those trivial outliers, although they achieve similar local affinity to the abnormal nodes, as shown in Fig. 3d, they are still not aligned well with the distribution of the anomaly nodes, and thus, they cannot serve as effective negative samples for learning the one-class classifier on the normal nodes. Thus, we further introduce an egocentric closeness prior-based loss function to tackle this issue, which models subtle abnormality on anomaly nodes, *i.e.*, the anomaly nodes that exhibit high similarity to the normal nodes.

More specifically, let $\mathbf{h}_i$ and $\hat{\mathbf{h}}_i$ be the representations of the normal node $v_i$ and its corresponding generated outlier node that shares the same ego network as $v_i$ (as discussed in Sec. 3.2), the egocentric closeness prior-based loss $\ell_{ec}$ is defined as follows:

$$\ell_{ec} = \frac{1}{|\mathcal{V}_o|} \sum_{v_i \in \mathcal{V}_o} \left\| \hat{\mathbf{h}}_i - (\mathbf{h}_i + \varepsilon) \right\|_2^2, \qquad (5)$$

where $|\mathcal{V}_o|$ is the number of the generated outliers and $\epsilon$ is a noise perturbation generated from a Gaussian distribution. The perturbation is added to guarantee a separability between $\mathbf{h}_i$ and $\hat{\mathbf{h}}_i$, while enforcing its egocentric closeness. It is worth mentioning that Gaussian noise works like a hyperparameter in the feature interpolation to diversify the outlier nodes in the feature representation space. Changes of this noise distribution do not affect the superiority of the performance of GGAD over the competing methods.

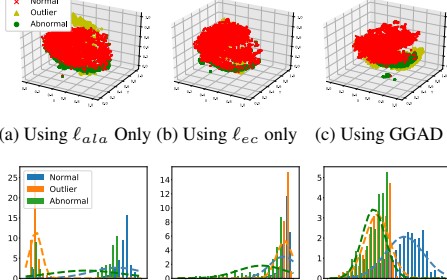

(a) Using $\ell_{ala}$ Only (b) Using $\ell_{ec}$ only (c) Using GGAD

(d) Using $\ell_{ala}$ Only (e) Using $\ell_{ec}$ Only (f) Using GGAD

Figure 3: (**a-c**) t-SNE visualization of the node representations and (**d-f**) histograms of local affinity yielded by GGAD and its two variants on a GAD dataset T-Finance [50].

As shown in Fig. 3c, using this egocentric closeness prior-based loss together with the local affinity prior-based loss learns outlier nodes that are well aligned to the real anomaly nodes in both the representation space and the local structure, as illustrated in Figs. 3c and 3f, respectively. Using the egocentric closeness alone also results in mismatches between the generated outlier nodes and the abnormal nodes (see Fig. 3e) since it ignores the local structure relation of the generated outlier nodes.

## 3.5 Graph Anomaly Detection using GGAD

**Training.** Since the generated outlier nodes are to assimilate the abnormal nodes, they can be used as important negative samples to train a one-class classifier on the labeled normal nodes. We implement this classifier using a fully connected layer on top of the GCN layers that maps the node representations to a prediction probability-based anomaly score, denoted by $\eta : \mathbf{H} \to \mathbb{R}$, followed by a binary cross-entropy (BCE) loss function $\ell_{bce}$:

$$\ell_{bce} = \sum_{i}^{|\mathcal{V}_o|+|\mathcal{V}_l|} y_i \log(p_i) + (1 - y_i) \log(1 - p_i), \qquad (6)$$

where $p_i = \eta(\mathbf{h}_i; \Theta_s)$ is the output of the one-class classifier indicating the probability that a node is a normal node, and $y$ is the label of node. We set $y = 1$ if the node is a labeled normal node, and $y = 0$ if the node is a generated outlier node. The one-class classifier is jointly optimized with the local affinity prior-based loss $\ell_{ala}$ and egocentric closeness prior-based loss $\ell_{ec}$ in an end-to-end manner. This results in mediation in the feature representation space where the generated outlier nodes are close to, yet separable from, the labeled normal nodes and their neighbors. Thus, these outlier nodes can be thought as hard anomalies that lie at the fringe of normal nodes in the feature representation space. The optimization of these two prior losses is continuously decreasing and converging finally during the training (see App. E), indicating that these two losses are collaborative rather than diverged. Thus, the overall loss $\ell_{total}$ can be formulated as:

$$\ell_{total} = \ell_{bce} + \beta \ell_{ala} + \lambda \ell_{ec}, \qquad (7)$$

where $\beta$ and $\lambda$ are the hyperparameters to control the importance of the two constraints respectively. The learnable parameters are $\Theta = \{\Theta_g, \Theta_s\}$.

**Inference.** During inference, we can directly use the inverse of the prediction of the one-class classifier as the anomaly score:

$$\text{score}(v_j) = 1 - \eta(\mathbf{h}_j; \Theta^*), \tag{8}$$

where $\Theta^*$ is the learned parameters of GGAD. Since our outlier nodes well assimilate the real abnormal nodes, they are expected to receive high anomaly scores from the one-class classifier.

## 4 Experiments

**Datasets.** We conduct experiments on six large real-world graph datasets with genuine anomalies from diverse domains, including the co-review network in Amazon [10], transaction record network in T-Finance [50], social networks in Reddit [21], bitcoin transaction in Elliptic [55], co-purchase network in Photo [35] and financial network in DGraph [18]. See App. A for more details about the datasets. Although it is easy to obtain normal nodes, the human checking and annotation of large-scale nodes are still costly. To simulate practical scenarios where we need to annotate only a relatively small number of normal nodes, we randomly sample $R\%$ of the normal nodes as labeled normal data for training, in which $R$ is chosen in $\{10, 15, 20, 25\}$, with the rest of nodes is treated as the testing set. Due to the massive set of nodes, the same $R$ applied to DGraph would lead to a significantly larger set of normal nodes than the other three data sets, leading to very different annotation costs in practice. Thus, on DGraph, $R$ is chosen in $\{0.05, 0.2, 0.35, 0.5\}$ to compose the training data.

**Competing Methods.** To our best knowledge, there exist no GAD methods specifically designed for semi-supervised node-level GAD. To validate the effectiveness of GGAD, we compare it with six state-of-the-art (SOTA) unsupervised methods and their advanced versions in which we effectively adapt them to our semi-supervised setting. These methods include two reconstruction-based models: DOMINANT [8] and AnomalyDAE [11], two one-class classification models: TAM [44] and OCGNN [53], and two generative models: AEGIS [7] and GAAN [6]. To effectively incorporate the normal information into these unsupervised methods, for the reconstruction models, DOMINANT and AnomalyDAE, the data reconstruction is performed on the labeled normal nodes only during training. In OCGNN, the one-class center is optimized based on the labeled normal nodes exclusively. In TAM, we train the model by maximizing the affinity on the normal nodes only. As for AEGIS and GAAN, the normal nodes combined with their generated outliers are used to train an adversarial classifier. Self-supervised-based methods like CoLA [28], SL-GAD [6], and HCM-A [17] and semi-supervised methods that require both labeled normal and abnormal nodes like GODM [25] and DiffAD [32] are omitted because training these methods on the data with exclusively normal nodes does not work.

**Evaluation Metric.** Following prior studies [5, 40, 51], two popular and complementary evaluation metrics for anomaly detection, the area under ROC curve (AUROC) and Area Under the precision-recall curve (AUPRC), are used to evaluate the performance. Higher AUROC/AUPRC indicates better performance. AUROC reflects the ability to recognize anomalies while at the same time considering the false positive rate. AUPRC focuses solely on the precision and recall rates of anomalies detected. The AUROC and AUPRC results are averaged over 5 runs with different random seeds.

**Implementation Details.** GGAD is implemented in Pytorch 1.6.0 with Python 3.7. and all the experiments are run on a 24-core CPU. In GGAD, its weight parameters are optimized using Adam [20] optimizer with a learning rate of $1e-3$ by default. For each dataset, the hyperparameters $\beta$ and $\lambda$ for two constraints are uniformly set to 1, though GGAD can perform stably with a range of $\beta$ and $\lambda$ (see App. C.2). The size of the generated outlier nodes $S$ is set to 5% of $|\mathcal{V}_l|$ by default and stated otherwise. The affinity margin $\alpha$ is set to 0.7 across all datasets. The perturbation in Eq. (5) is drawn from a Gaussian distribution, with mean and standard variance set to 0.02 and 0.01 respectively, and it is stated otherwise. All the competing methods are implemented by using their publicly available official source code or the library, and they are trained using their suggested hyperparameters. To apply GGAD and the competing models to very large graph datasets, *i.e.*, DGraph, a min-batch training strategy is applied (see Algorithm 2 for detail).

Table 1: AUROC and AUPRC on six GAD datasets. The best performance per dataset is boldfaced, with the second-best underlined. '/' indicates that the model cannot handle the DGraph dataset.

| Setting | Method | Dataset | | | | | | | | | | | |
|---|---|---|---|---|---|---|---|---|---|---|---|---|---|
| | | AUROC | | | | | | AUPRC | | | | | |
| | | Amazon | T-Finance | Reddit | Elliptic | Photo | DGraph | Amazon | T-Finance | Reddit | Elliptic | Photo | DGraph |
| Unsupervised | DOMINANT | 0.7025 | 0.6087 | 0.5105 | 0.2960 | 0.5136 | 0.5738 | 0.1315 | 0.0536 | 0.0380 | 0.0454 | 0.1039 | 0.0075 |
| | AnomalyDAE | 0.7783 | 0.5809 | 0.5091 | 0.4963 | 0.5069 | 0.5763 | 0.1429 | 0.0491 | 0.0319 | 0.0872 | 0.0987 | 0.0070 |
| | OCGNN | 0.7165 | 0.4732 | 0.5246 | 0.2581 | 0.5307 | / | 0.1352 | 0.0392 | 0.0375 | 0.0616 | 0.0965 | / |
| | AEGIS | 0.6059 | 0.6496 | 0.5349 | 0.4553 | 0.5516 | 0.4509 | 0.1200 | 0.0622 | 0.0413 | 0.0827 | 0.0972 | 0.0053 |
| | GAAN | 0.6513 | 0.3091 | 0.5216 | 0.2590 | 0.4296 | / | 0.0852 | 0.0283 | 0.0348 | 0.0436 | 0.0767 | / |
| | TAM | 0.8303 | 0.6175 | 0.6062 | 0.4039 | 0.5675 | / | 0.4024 | 0.0547 | 0.0437 | 0.0502 | 0.1013 | / |
| Semi-supervised | DOMINANT | 0.8867 | 0.6167 | 0.5194 | 0.3256 | 0.5314 | 0.5851 | 0.7289 | 0.0542 | 0.0414 | 0.0652 | 0.1283 | 0.0076 |
| | AnomalyDAE | 0.9171 | 0.6027 | 0.5280 | 0.5409 | 0.5272 | 0.5866 | 0.7748 | 0.0538 | 0.0362 | 0.0949 | 0.1177 | 0.0071 |
| | OCGNN | 0.8810 | 0.5742 | 0.5622 | 0.2881 | 0.6461 | / | 0.7538 | 0.0492 | 0.0400 | 0.0640 | 0.1501 | / |
| | AEGIS | 0.7593 | 0.6728 | 0.5605 | 0.5132 | 0.5936 | 0.4450 | 0.2616 | 0.0685 | 0.0441 | 0.0912 | 0.1110 | 0.0058 |
| | GAAN | 0.6531 | 0.3636 | 0.5349 | 0.2724 | 0.4355 | / | 0.0856 | 0.0324 | 0.0362 | 0.0611 | 0.0768 | / |
| | TAM | 0.8405 | 0.5923 | 0.5829 | 0.4150 | 0.6013 | / | 0.5183 | 0.0551 | 0.0446 | 0.0552 | 0.1087 | / |
| | **GGAD** (Ours) | **0.9443** | **0.8228** | **0.6354** | **0.7290** | **0.6476** | **0.5943** | **0.7922** | **0.1825** | **0.0610** | **0.2425** | 0.1442 | **0.0082** |

## 4.1 Main Comparison Results

Table 1 shows the comparison of GGAD to 12 models, in which semi-supervised models use 15% normal nodes during training while unsupervised methods are trained on the full graph in a fully unsupervised way. We will discuss results using more/less training normal nodes in Sec. 4.2.

**Comparison to Unsupervised GAD Methods**. As shown in Table 1, GGAD significantly outperforms all unsupervised methods on six datasets, having maximally 21% AUROC and 39% AUPRC improvement over the best-competing unsupervised methods on individual datasets. The results also show that the semi-supervised versions of the unsupervised methods largely improve the performance of their unsupervised counterparts, justifying that i) incorporating the normal information into the unsupervised approaches is beneficial for enhancing the detection performance and ii) our approach to adapt the unsupervised methods is effective across various types of GAD models. TAM performs best among the unsupervised methods. AEGIS which leverages GAN to learn the normal patterns performs better than AnomalyDAE and DOMINANT on T-Finance, Reddit, and Photo, By contrast, reconstruction-based methods work well on Amazon and DGraph. Similar observations can be found for the semi-supervised versions.

**Comparison to Semi-supervised GAD Methods**. The results in Table 1 show that although the semi-supervised methods largely outperform unsupervised counterparts, they substantially underperform our method GGAD. The reconstruction-based approaches show the most competitive performance among the contenders in semi-supervised settings, *e.g.*, AnomalyDAE performs best on Amazon and DGraph. Nevertheless, GGAD gains respectively about 1-3% AUROC/AUPRC improvement on these two datasets compared to best-competing AnomalyDAE.

By training on the normal nodes only, methods like TAM and AEGIS largely reduce the interference of unlabeled anomaly nodes on the model and work well on most of the datasets, *e.g.*, TAM on Amazon and Reddit, AEGIS on T-Finance and Reddit. However, their performance is still lower than GGAD by a relatively large margin. GGAD yields the best AUROC on the Photo while yielding the second-best in AUPRC, underperforming OCGNN. This indicates that GGAD can detect some anomalies very accurately in Photo, but it is less effective than OCGNN to get a bit more anomalies rank at the top of normal nodes in terms of their anomaly score. On average over the six datasets, GGAD outperforms the best semi-supervised contender AnomalyDAE by 11% in AUROC and 5% in AUPRC, demonstrating that GGAD can make much better use of the labeled normal nodes through our two anomaly prior-based losses.

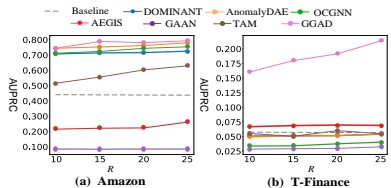

Figure 4: AUPRC results w.r.t the size of training normal nodes ($R$% of $|\mathcal{V}|$). 'Baseline' denotes the performance of the best unsupervised GAD method.

## 4.2 Performance w.r.t. Training Size and Anomaly Contamination

In order to further illustrate the effectiveness of our method, we also compare GGAD with other semi-supervised methods using varying numbers of training normal nodes in Fig. 4 and having

various anomaly contamination rates in Fig. 5. Due to page limitation, we present the AUPRC results on two datasets here only, showing the representative performance. The full AUROC and AUPRC results are reported in App. C.

The results in Fig. 4 show that with increasing training samples of normal nodes, the performance of all methods on all four datasets generally gets improved since more normal samples can help the models more accurately capture the normal patterns during training. Importantly, GGAD consistently outperforms all competing methods with varying numbers of normal nodes, reinforcing that GGAD can make better use of the labeled normal nodes for GAD.

The labeled normal nodes can often be contaminated by anomalies due to factors like annotation errors. To consider this issue, we introduce a certain ratio of anomaly contamination into into the training normal node set $\mathcal{V}_l$. The results of the models under different ratios of contam-

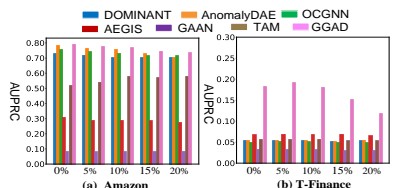

Figure 5: AUPRC w.r.t. contamination.

ination in Fig. 5. show that with increasing anomaly contamination, the performance of all methods decreases. Despite the decreased performance, our method GGAD consistently maintains the best performance under different contamination rates, showing good robustness w.r.t. the contamination.

## 4.3 Ablation Study

**Importance of the Two Anomaly Node Priors.** The importance of the two proposed losses based on the priors on the anomaly nodes is examined by comparing our full model with its variant removing the corresponding loss, with the results shown in Table 4. It is clear that learning the outlier node representations using one of the two losses performs remarkably less effectively than our full model using both losses.

It is mainly because although using $\ell_{ala}$ solely can obtain similar local affinity of the outliers to the real anomaly nodes, the outliers are still not aligned well with the distribution of the anomaly nodes in the node representation space. Likewise, only using the $\ell_{ec}$ can result in a mismatch between the generated outliers and real abnormal samples in their graph structure. GGAD that effectively unifies both priors through the two losses can generate outlier nodes that well assimilate the real abnormal nodes on both graph structure and node representation space, supporting substantially more accurate GAD performance.

Table 2: Ablation study on our two priors.

| Metric | Component | | Dataset | | | | | |
|---|---|---|---|---|---|---|---|---|
| | $\ell_{ala}$ | $\ell_{ec}$ | Amazon | T-Finance | Reddit | Elliptic | Photo | DGraph |
| AUROC | | ✓ | 0.8871 | 0.8149 | 0.5839 | 0.6863 | 0.5762 | 0.5891 |
| | ✓ | | 0.7250 | 0.6994 | 0.5230 | 0.7001 | 0.6103 | 0.5513 |
| | ✓ | ✓ | **0.9324** | **0.8228** | **0.6354** | **0.7290** | **0.6476** | **0.5943** |
| AUPRC | | ✓ | 0.6643 | 0.1739 | 0.0409 | 0.1954 | 0.1137 | 0.0076 |
| | ✓ | | 0.1783 | 0.0800 | 0.0398 | **0.2683** | 0.1186 | 0.0063 |
| | ✓ | ✓ | **0.7843** | **0.1924** | **0.0610** | 0.2425 | **0.1442** | **0.0087** |

**GGAD vs. Alternative Outlier Node Generation Approaches.** To examine its effectiveness further, GGAD is also compared with four other approaches that could be used as an alternative to generating the outlier nodes. These include (i) **Random** that randomly sample some normal nodes and treat them as outliers to train a one-class discriminative classifier, (ii) **Non-learnable Outliers (NLO)** that removes the learnable parameters $\tilde{\mathbf{W}}$ in Eq. (2) in our outlier node generation, (iii) **Noise** that directly generates the representation of outlier nodes from random noise, (iv) **Gaussian Perturbation (GaussianP)** that directly adds Gaussian perturbations into the sampled normal nodes' representations

Table 3: GGAD vs. alternative outlier generators.

| Metric | Method | Dataset | | | | | |
|---|---|---|---|---|---|---|---|
| | | Amazon | T-Finance | Reddit | Elliptic | Photo | DGraph |
| AUROC | Random | 0.7263 | 0.4613 | 0.5227 | 0.6856 | 0.5678 | 0.5712 |
| | NLO | 0.8613 | 0.6179 | 0.5638 | 0.6787 | 0.5307 | 0.5538 |
| | Noise | 0.8508 | 0.8204 | 0.5285 | 0.6786 | 0.5940 | 0.5779 |
| | GaussianP | 0.2279 | 0.6659 | 0.5235 | 0.6715 | 0.5925 | 0.5862 |
| | VAE | 0.8984 | 0.6674 | 0.6175 | 0.7055 | 0.6222 | 0.5801 |
| | GAN | 0.8288 | 0.5487 | 0.5378 | 0.6256 | 0.6032 | 0.5101 |
| | **GGAD (Ours)** | **0.9324** | **0.8228** | **0.6354** | **0.7290** | **0.6476** | **0.5943** |
| AUPRC | Random | 0.1755 | 0.0402 | 0.0394 | 0.1981 | 0.1063 | 0.0061 |
| | NLO | 0.4696 | 0.1364 | 0.0495 | 0.1750 | 0.1092 | 0.0065 |
| | Noise | 0.5384 | 0.1762 | 0.0381 | 0.1924 | 0.1200 | 0.0076 |
| | GaussianP | 0.0397 | 0.0677 | 0.0376 | 0.1682 | 0.1194 | 0.0078 |
| | VAE | 0.6111 | 0.0652 | 0.0528 | 0.2344 | 0.1272 | 0.0063 |
| | GAN | 0.3715 | 0.0461 | 0.0433 | 0.1263 | 0.1143 | 0.0051 |
| | **GGAD (Ours)** | **0.7843** | **0.1924** | **0.0610** | **0.2425** | **0.1442** | **0.0087** |

to generate the outliers. Apart from the **Noise** and **GaussianP**, we further employ two advanced generation approaches, (vi) **VAE** that generate the outlier representations by reconstructing the raw attributes of selected nodes where our two anomaly prior-based constraints are applied to the generation, and (v) **GAN** that generates the embedding from the noise and adds an adversarial function to discriminate whether the generated node is fake or real, with our two prior constraints applied in

the generation as well. The results are shown in Table 3. **Random** does not work properly, since the randomly selected samples are not distinguishable from the normal nodes. **NLO** performs fairly well on some data sets such as Amazon, T-Finance, and Elliptic, but it is still much lower than GGAD, showcasing that having learnable outlier node representations can help better ground the outliers in a real local graph structure. Despite that **Noise** and **GaussianP** can generate outliers that have separable representations from the normal nodes, they also fail to work well since the lack of graph structure in the outlier nodes can lead to largely mismatched distributions between the generated outlier nodes and the anomaly nodes. By contrast, the outlier nodes learned by GGAD can better align with the anomaly nodes due to the incorporation of the anomaly priors on graph structure and feature representation into our GAD modeling. Both **VAE** and **GAN** can work well on some datasets, which indicates two priors help them learn relevant outlier representations. But both of them are still much lower than GGAD, showcasing that the outlier generation approach in GGAD can leverage the two proposed priors to generate better outliers.

**GGAD vs. GGAD enabled Unsupervised Methods.** We incorporate the outlier generation into existing unsupervised methods to demonstrate the generation in GGAD can also benefit the existing unsupervised methods. To allow the unsupervised methods to fully exploit the generated outliers, we first utilize GGAD to generate outlier nodes by training on randomly sampled nodes from a graph (which can be roughly treated as all normal nodes due to anomaly scarcity) and then remove possible abnormal nodes from the graph dataset by filtering out Top-K

most similar nodes to the generated outlier nodes. By removing these suspicious abnormal nodes, the unsupervised method is expected to train on the cleaner graph (*i.e.*, with less anomaly contamination). This approach to improve unsupervised GAD methods is referred to as GGAD-enabled unsupervised GAD. We evaluate their effectiveness on three large-scale datasets. As shown in Table 4, where #Anomalies/#Top-K Node represents the number of real abnormal nodes we successfully filter out and the number of nodes we choose to filter out (*i.e.*, K) respectively. For example, we use the outlier nodes generated by GGAD to filter out 500 nodes from the Amazon dataset, of which there are 387 real

Table 4: GGAD enabled unsupervised methods.

| Metric | Method | Dataset | | |
|---|---|---|---|---|
| | | Amazon | T-Finance | Elliptic |
| #Anomalies/#Top-K Nodes | | 387/500 | 351/1000 | 1448/2000 |
| AUROC | DOMINANT | 0.7025 | 0.6087 | 0.2960 |
| | GGAD-enabled DOMINANT | 0.8186 | 0.6275 | 0.2986 |
| | OCGNN | 0.7165 | 0.4732 | 0.2581 |
| | GGAD-enabled OCGNN | 0.8692 | 0.5931 | 0.2638 |
| | AEGIS | 0.6059 | 0.6496 | 0.4553 |
| | GGAD-enabled AEGIS | 0.8395 | 0.7024 | 0.5036 |
| | GGAD | **0.9431** | **0.8108** | **0.7225** |
| AUPRC | DOMINANT | 0.1315 | 0.0536 | 0.0454 |
| | GGAD-enabled DOMINANT | 0.3462 | 0.0585 | 0.0613 |
| | OCGNN | 0.1352 | 0.0392 | 0.0616 |
| | GGAD-enabled OCGNN | 0.3950 | 0.0480 | 0.0607 |
| | AEGIS | 0.1200 | 0.0622 | 0.0827 |
| | GGAD-enabled AEGIS | 0.3833 | 0.0784 | 0.0910 |
| | GGAD | **0.7769** | **0.1734** | **0.2484** |

abnormal nodes. This helps largely reduce the anomaly contamination rate in the graph. The results show that this approach can significantly improve the performance of three different representative unsupervised GAD methods, including DOMINANT, OCGNN, and AEGIS. Note that although the GGAD-enabled unsupervised methods achieve better performance, their performance still largely underperforms GGAD, which provides stronger evidence for the effective capability in anomaly detection of GGAD.

## 5 Conclusion and Future Work

In this paper, we investigate a new semi-supervised GAD scenario where part of normal nodes are known during training. To fully exploit those normal nodes, we introduce a novel outlier generation approach GGAD that leverages two important priors about anomalies in the graph to learn outlier nodes that well assimilate real anomalies in both graph structure and feature representation space. The quality of these outlier nodes is justified by their effectiveness in training a discriminative one-class classifier together with the given normal nodes. Comprehensive experiments are performed to establish an evaluation benchmark on six real-world datasets for semi-supervised GAD, in which our GGAD outperforms 12 competing methods across the six datasets.

**Limitation and Future work.** The generation of the outlier nodes in GGAD is built upon the two important priors about anomaly nodes in a graph. This helps generate outlier nodes that well assimilate the anomaly nodes across diverse real-world GAD datasets. However, these priors are not exhaustive, and there can be some anomalies whose characteristics may not be captured by the two priors used. We will explore this possibility and improve GGAD for this case in our future work.

**Acknowledgments.** We thank the anonymous reviewers for their valuable comments. The participation of Guansong Pang was supported in part by Lee Kong Chian Fellowship.

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

# A   Detailed Dataset Description

The key statistics of the datasets are presented in Table 5. A detailed introduction of these datasets is given as follows.

- Amazon [10]: It includes product reviews under the Musical Instrument category. The users with more than 80% of helpful votes were labeled as begin entities, with the users with less than 20% of helpful votes treated as fraudulent entities. There are three relations including U-P-U (users reviewing at least one same product), U-S-U (users giving at least one same star rating within one week), and U-V-U (users with top-5% mutual review similarities). In this paper, we do not distinguish this connection and regard them as the same type of edges, *i.e.*, all connections are used. There are 25 handcrafted features that were collected as the raw node features.

- T-Finance [50]: It is a financial transaction network where the node represents an anonymous account and the edge represents two accounts that have transaction records. The features of each account are related to some attributes of logging, such as registration days, logging activities, and interaction frequency, etc. The users are labeled as anomalies when they fall into the categories of fraud money laundering and online gambling.

- Reddit [21]: It is a user-subreddit graph, capturing one month's worth of posts shared across various subreddits at Reddit. The users who have been banned by the platform are labeled anomalies. The text of each post is transformed into the feature vector and the features of the user and subreddits are the feature summation of the post they have posted.

- Elliptic [55]: It is a bitcoin transaction network in which the node represents the transactions and the edge is the flow of Bitcoin currency.

- Photo [35]: It is an Amazon co-purchase network in which the node represents the product and the edge represents the co-purchase relationship. The attribute of the node is a bag of works representation of the user's comments.

- DGraph [18]: It is a large-scale attributed graph with millions of nodes and edges where the node represents a user account in a financial company and the edge represents that the user was added to another account as an emergency contact. The feature of a node is the profile information of users, such as age, gender, and other demographic features. The users who have overdue history are labeled as anomalies.

# B   More Information about the Competing Methods

## B.1   Competing Methods

A more detailed introduction of the six GAD models we compare with is given as follows.

- DOMINANT [8] leverages the auto-encoder for graph anomaly detection. It consists of an encoder layer and a decoder layer which are devised to reconstruct the features and structure of the graph. The reconstruction errors from the features and the structural modules are combined as an anomaly score.

- AnomalyDAE [11] consists of a structure autoencoder and an attribute autoencoder to learn both node embeddings and attribute embeddings jointly in a latent space. In addition, an attention mechanism is employed in the structure encoder to capture normal structural patterns more effectively.

- OCGNN [53] applies one-class SVM and GNNs, aiming at combining the recognition ability of one-class classifiers and the powerful representation of GNNs. A one-class hypersphere learning objective is used to drive the training of the GNN. The sample that falls outside the hypersphere is defined as an anomaly.

- AEGIS [7] designs a new graph neural layer to learn anomaly-aware node representations and further employ generative adversarial networks to detect anomalies among new data. The generator takes noises sampled from a prior distribution as input, and attempts to generate informative pseudo anomalies. The discriminator tries to distinguish whether an input is the representation of a normal node or a generated anomaly.

Table 5: Key statistics of the six datasets used in our experiments.

| Dataset | Type | # Nodes | # Edges | # Attributes | #Anomalies (Rate) |
|---------|------|---------|---------|--------------|-------------------|
| Amazon | Co-review | 11,944 | 4,398,392 | 25 | 821(6.9%) |
| T-Finance | Transaction | 39,357 | 21,222,543 | 10 | 1,803(4.6%) |
| Reddit | Social Media | 10,984 | 168,016 | 64 | 366(3.3%) |
| Elliptic | Bitcoin Transaction | 46,564 | 73,248 | 93 | 4,545 (9.8%) |
| Photo | Co-purchase | 7,535 | 119,043 | 745 | 698(9.2%) |
| DGraph | Financial Networks | 3,700,550 | 73,105,508 | 17 | 15,509(1.3%) |

- GAAN [6] is based on a generative adversarial network where fake graph nodes are generated by a generator. To encode the nodes, they compute the sample covariance matrix for real nodes and fake nodes, and a discriminator is trained to recognize whether two connected nodes are from a real or fake node.

- TAM [44] learns tailored node representations for a local affinity-based anomaly measure by maximizing the local affinity of nodes to their neighbors. TAM is optimized on truncated graphs where non-homophily edges are removed iteratively. The learned representations result in significantly stronger local affinity for normal nodes than abnormal nodes. So, the local affinity of a node in the learned representation space is used as anomaly score.

### B.2 Official Source Code.

All the competing methods except TAM are implemented by PyGOD Library [23, 24]. The code of TAM is taken from its authors. The links to their source codes are as follows:

- PyGOD: https://github.com/pygod-team/pygod
- TAM: https://github.com/mala-lab/TAM-master
- AEGIS: https://github.com/pygod-team/pygod
- GAAN: https://github.com/pygod-team/pygod
- DOMINANT: https://github.com/kaize0409/GCN_AnomalyDetection_pytorch
- AnomalyDAE: https://github.com/haoyfan/AnomalyDAE
- OCGNN: https://github.com/WangXuhongCN/OCGNN

## C  Additional Experimental Results

### C.1  More Prior Information

To further verify asymmetric local affinity, we provide more affinity visualization results on other GAD datasets including Amazon, Reddit, Elliptic, and Photo, as shown in Fig. 6. The results show that the normal nodes have a much stronger affinity to their neighboring normal node than the sampled abnormal nodes.

For the egocentric closeness prior, the feature representations of outlier nodes should be closed to the normal nodes that share similar local structure as the outlier nodes, we verify this prior by analyzing the similarity between normal and abnormal nodes based on the raw node attributes on the other four datasets shown in Fig. 7. The results show that the real abnormal nodes can exhibit high similarity to the normal nodes in terms of local affinity in the raw attribute space. The main reason is that some abnormalities are weak or the existence of adversarial camouflage that disguises abnormal nodes to have similar attributes to the local community. This is the key intuition behind the second prior.

### C.2  Sensitivity Analysis

This section analyzes the sensitivity of GGAD w.r.t four key hyperparameters, including the affinity margin $\alpha$, hyperparameters of structural affinity loss $\beta$ and egocentric closeness $\lambda$, and the number of generated outlier nodes $S$. The AUPRC results are reported in Fig. 8. We discuss these results below.

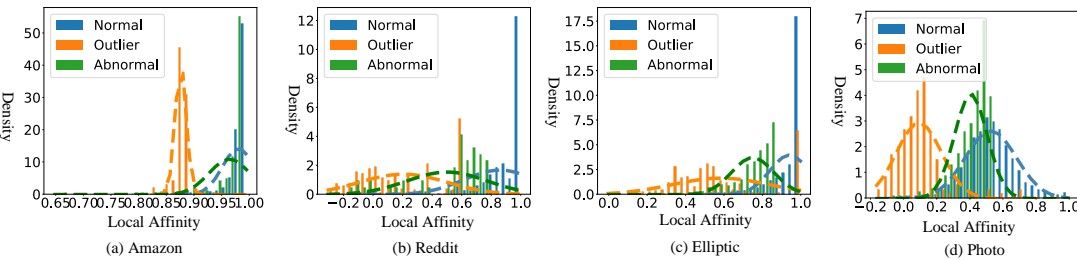

Figure 6: Histogram of local affinity on more datasets

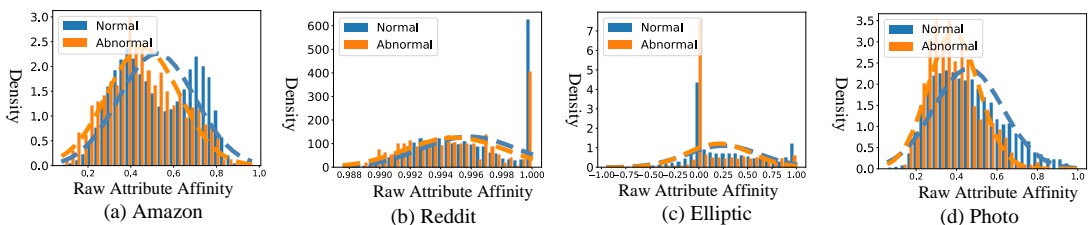

Figure 7: Affinity distribution based on raw node attributes

**Impact of Margin** $\alpha$ **in** $\ell_{ala}$. $\alpha$ in $\ell_{ala}$ denotes the affinity difference we enforce between the normal and outlier nodes. As Fig. 8(a) shows, GGAD performs best on Amazon and Photo with increasing $\alpha$ while it performs stably on the other four datasets with varying $\alpha$, indicating that enforcing local separability is more effective on Amazon and Photo than the other datasets, as can also be observed in Table 3.

**Impact of Hyperparameters** $\beta$ **and** $\lambda$. As shown in Figs. 8(b)(c), with increasing $\beta$ and $\lambda$, our model GGAD generally performs better, indicating that a stronger structural affinity or egocentric closeness constraint is generally preferred to generate outlier nodes that are more aligned with the real abnormal nodes.

**The Number of Generated Outlier Nodes** $S$. As shown in Fig. 8(d), where $S$ indicates that we generate the outlier nodes in a quantity at a rate of $S\%$ of $|\mathcal{V}_l|$, GGAD gains some improvement with more generated outlier nodes but it maintains the same performance after a certain number of outlier nodes. This is mainly because the generated outlier nodes may not be diversified enough to resemble all types of abnormal nodes, even with much more outlier nodes. The declined performance on Amazon and Photo is mainly due to the fact that the labeled normal data in these two datasets is small and is overwhelmed by increasing outlier nodes, leading to worse training of the one-class classifier. The AUROC results of these four key hyperparameters are shown in Fig. 9, which show a similar trend as the AUPRC results.

### C.3 More Results for Models Trained on Varying Number of Normal Nodes

The results of AUPRC and AUROC under different training normal sample sizes are shown in Fig. 10. and Fig. 11 respectively. The results show that increasing training samples of normal nodes can help the methods more accurately capture the normality, resulting in a consistent improvement. Among these methods, GGAD consistently outperforms the competing methods with varying numbers of training normal nodes, indicating that GGAD can make better use of labeled normal nodes.

### C.4 More Results for Robustness w.r.t. Anomaly Contamination

The full AUROC and AUPRC results under different anomaly contamination rates for all six real-world datasets are shown in Fig. 12 and Fig. 13, respectively. The results show that the performance of all methods decreases with the increasing rate of contamination. The reconstruction-based methods DOMINANT and AnomalyDAE are the most sensitive models, followed by GAAN, TAM, and AEGIS. OCGNN is relatively stable under the contamination. Despite the declined performance for

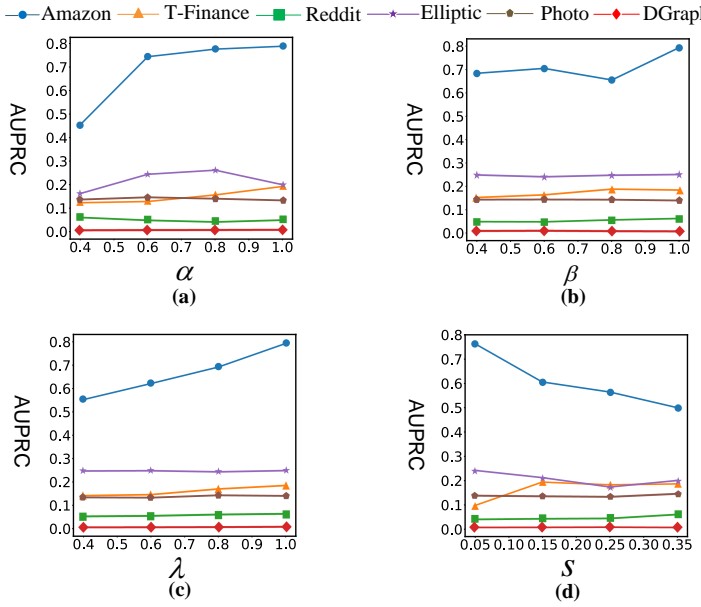

Figure 8: AUPRC of GGAD w.r.t hyperparameters $\alpha$, $\beta$, $\lambda$, $S$.

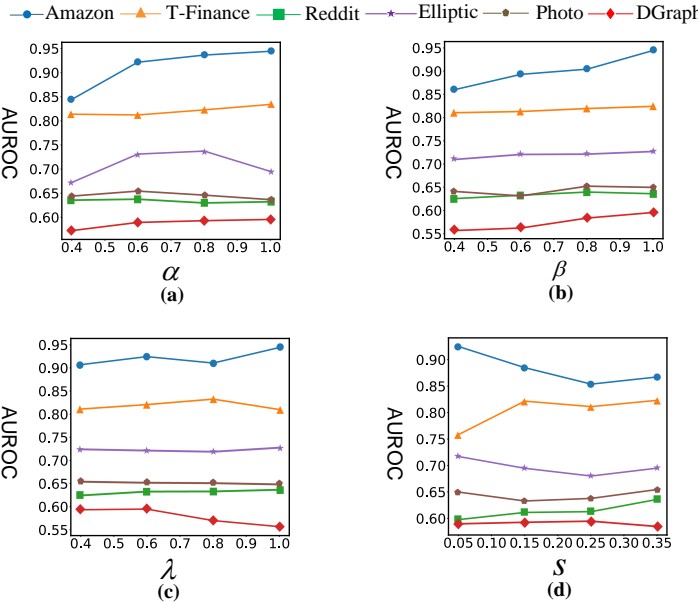

Figure 9: AUROC results w.r.t hyperparameters $\alpha$, $\beta$, $\lambda$, $S$.

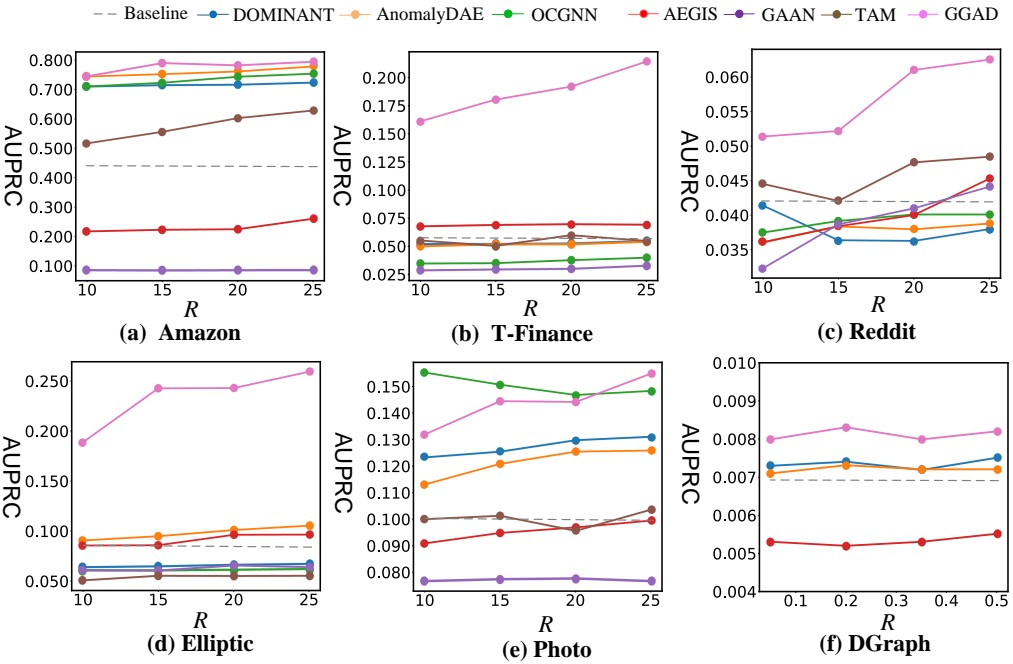

Figure 10: AUPRC results w.r.t. different number of training normal nodes ($R\%$ of $|\mathcal{V}|$). DGraph is a very large dataset, so a significantly smaller $R$ is used to have a similar size of $\mathcal{V}_l$ as the other datasets. 'Baseline' denotes the performance of the best unsupervised GAD method per dataset.

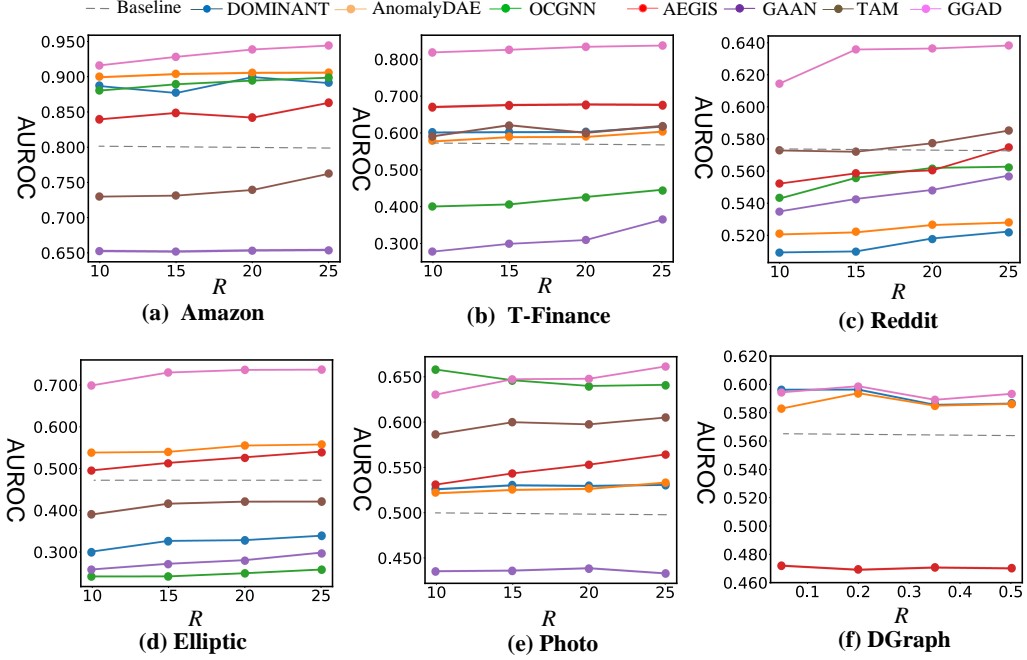

Figure 11: AUROC results w.r.t. different number of training normal nodes ($R\%$ of $|\mathcal{V}|$). DGraph is a very large dataset, so a significantly smaller $R$ is used to have a similar size of $\mathcal{V}_l$ as the other datasets. 'Baseline' denotes the performance of the best unsupervised GAD method per dataset.

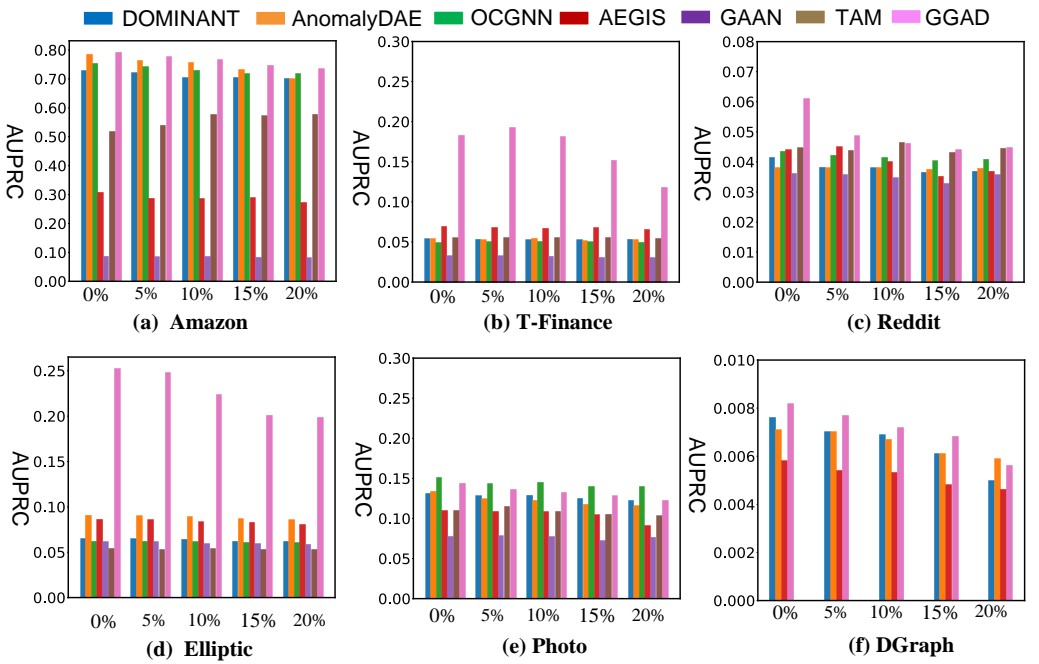

Figure 12: AUPRC w.r.t. different anomaly contamination.

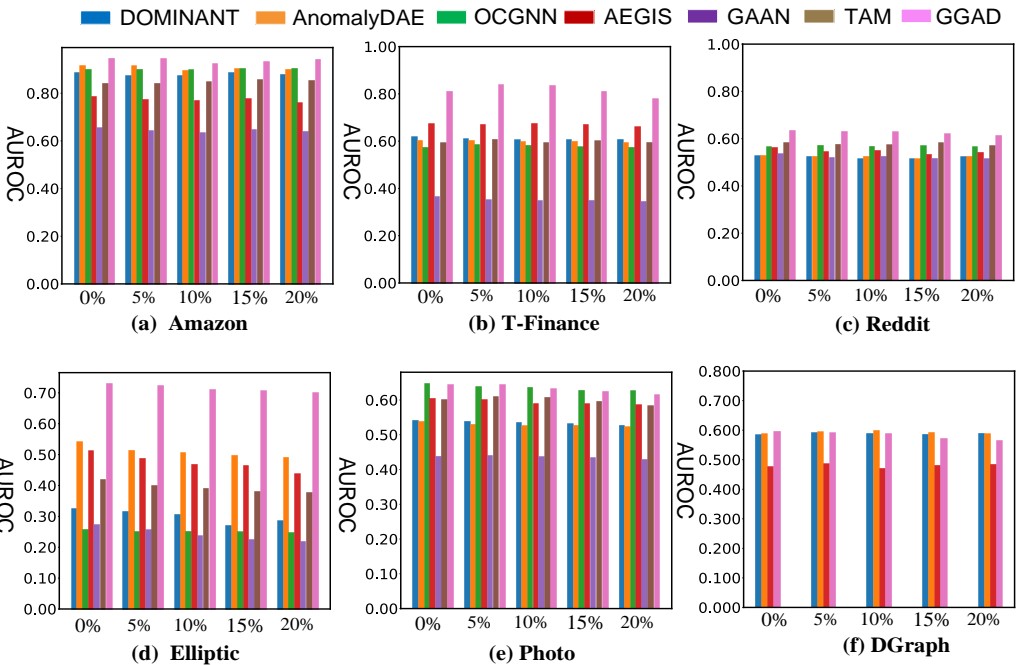

Figure 13: AUROC w.r.t. different anomaly contamination.

all the methods, our method GGAD still maintains the best performance under different contamination rates.

## C.5 More Analysis on the Generated Outliers

We further employ the Maximum Mean Discrepancy (MMD) distance to measure the distance between the generated outliers and the real abnormal nodes (and the normal data as well) to illustrate more in-depth characteristics of the generated outlier nodes. As shown in Table. 6, it is clear that the

Table 6: Analysis of the generated outlier nodes using MMD distance.

| Metric | Dataset | | | | |
| --- | --- | --- | --- | --- | --- |
| | Amazon | T-Finance | Elliptic | Photo | Reddit |
| with Abnormal Node | **0.1980** | **0.0784** | **0.1094** | **0.3703** | **0.3409** |
| with Normal Node | 0.2318 | 0.1040 | 0.1304 | 0.3880 | 0.3605 |

Table 7: Runtimes (in seconds) on the six datasets on CPU.

| Method | Dataset | | | | | |
| --- | --- | --- | --- | --- | --- | --- |
| | Amazon | T-Finance | Reddit | Elliptic | Photo | DGraph |
| DOMINANT | 1592 | 10721 | 125 | 1119 | 437 | 388 |
| AnomalyDAE | 1656 | 18560 | 161 | 8296 | 445 | 457 |
| OCGNN | 765 | 5717 | 162 | 3517 | 125 | / |
| AEGIS | 1121 | 15258 | 166 | 5638 | 417 | 1022 |
| GAAN | 1678 | 12120 | 94 | 1866 | 307 | / |
| TAM | 4516 | 17360 | 432 | 13200 | 165 | / |
| **GGAD** (Ours) | 658 | 9345 | 368 | 5146 | 106 | 488 |

distribution of the generated outliers have much smaller MMD distance to the real abnormal nodes than the normal nodes, indicating the good alignment of the distribution of the generated outliers with the real abnormal nodes.

## D   Computational Efficiency Analysis

### D.1   Time Complexity Analysis

This subsection analyzes the time complexity of GGAD. We build a GCN to obtain the representation of each node, which takes $O(mdh)$, where $m$ is the number of non-zero elements in matrix $\mathbf{A}$, $d$ is the dimension of representation, and $h$ is the number of feature maps. The outliers are generated from the ego network of a labeled normal node, which takes $O(Skd^2)$ where $S$ is the number of generated outliers and $k$ is the number of average neighbors for each outlier. The affinity calculation will take $O(N^2d)$, where $N$ is the number of nodes. The structural affinity and egocentric closeness losses take $O(N)$ and $O(Sd)$, respectively. The MLP layer mapping the representation to the anomaly score takes $O(Nd^2)$. Thus, the overall complexity of GGAD is $O(mdh + Skd^2 + N^2d + N + Sd + Nd^2)$.

### D.2   Runtime Results

The runtimes, including both training and inference time, of GGAD and six semi-supervised competing methods on CPU are shown in Table 7. In GGAD, although calculating the local affinity of each node requires some overheads, it is still much more efficient than the reconstruction operations on both the attributes and the structure as in DOMINANT and AnomalyDAE. Compared to the generative models AEGIS and GAAN, GGAD is generally more efficient on larger graph datasets like Amazon, T-Finance, and DGraph. OCGNN is a model with the simplest operation, to which our GGAD can also have comparable efficiency. These results demonstrate the advantage of GGAD in computational efficiency

## E   The Training Curves of Optimization

To further demonstrate the collaboration between these two prior-based losses, we visualize the optimization of loss during the training in Fig. 14, where 'ala' and 'ec' represent the two priors losses and the 'total' represents the sum of these two priors and the BCE loss. From the results, we can see that the two prior losses and the total loss are continuously decreasing and converging finally, indicating that these optimizations are collaborative rather than diverged.

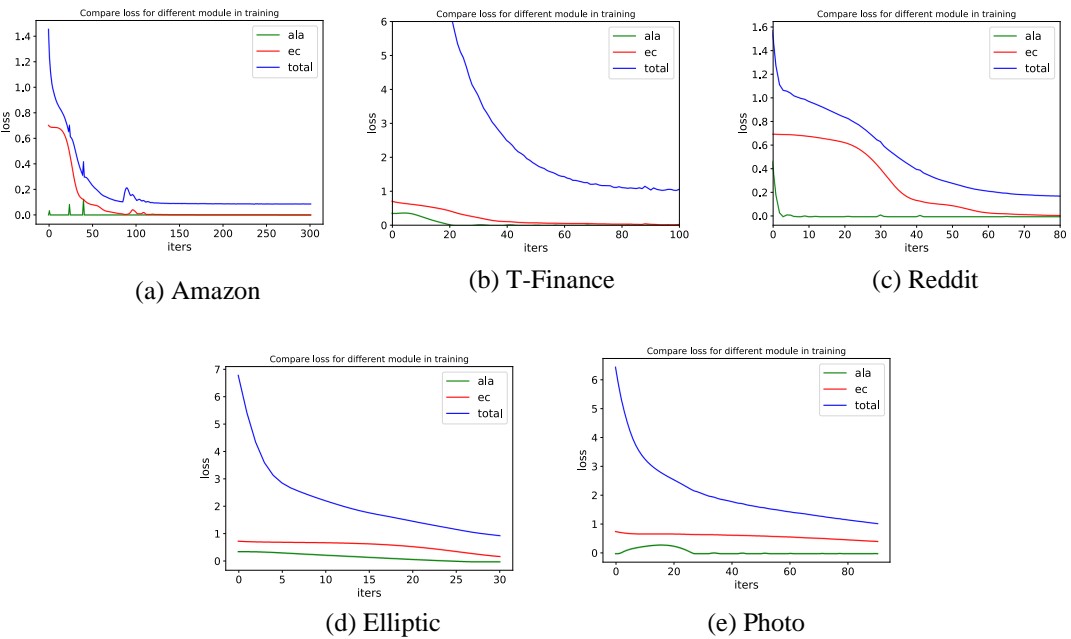

(a) Amazon

(b) T-Finance

(c) Reddit

(d) Elliptic

(e) Photo

Figure 14: Loss curve of different modules in GGAD

# F    Pseudo Code of GGAD

The training algorithms of GGAD are summarized in Algorithm 1 and Algorithm 2. Algorithm 1 describes the full training process of GGAD. Algorithm 2 describes the mini-batch processing for handling very large graph datasets, *i.e.*, DGraph. Since the number of the outlier nodes is significantly smaller than that of the normal nodes, we guarantee that each mini-batch consists of both normal and outlier nodes to address the data imbalance problem. The outputs are the mini-batches of samples from the graph and corresponding structural information, which can then be used as the input of GGAD or the competing models to perform GAD on DGraph. Note that when using Algorithm 2 for the competing models, the steps that involve the generated outliers are not used if they do not have the outlier generation component.

---

**Algorithm 1** GGAD

---

**Input**: Graph, $\mathcal{G} = (\mathcal{V}, \mathcal{E}, \mathbf{X})$, $N$: Number of training nodes, , $N_u$: Number of unlabeled nodes, $L$: Number of layers, $E$: Training epochs, $\mathcal{V}_{train}$:Training set, $\mathcal{V}_{test}$:Test set , $S$: The number of the generated outlier nodes
   **Output**: Anomaly scores of all nodes.

1: Sample the ego networks of some normal nodes upon which the outlier nodes are to be generated $\mathcal{V}_o = [v_{o_1}, ..., v_{o_s}]$
2: Compose $\mathcal{V}_{train}$ with the outlier nodes $\mathcal{V}_o$ and the given labeled normal node set $\mathcal{V}_l$
3: Randomly initialize GNN $(\mathbf{h}_1^{(0)}, \mathbf{h}_2^{(0)}, ..., \mathbf{h}_N^{(0)}) \leftarrow \mathbf{X}$
4: **for** $epoch = 1, \cdots, E$ **do**
5:    **for** each $v$ in $\mathcal{V}_{train}$ **do**
6:       **for** $l = 1, \cdots, L$ **do**
7:          $\mathbf{h}_v^{(l)} = \phi(\mathbf{h}_v^{(l-1)}; \mathbf{\Theta})$
8:          $\mathbf{h}_v^{(l)} = \mathrm{ReLU}\left(\mathrm{AGG}(\{\mathbf{h}_{v'}^{(l)} : (v, v') \in \mathcal{E}\})\right)$
9:       **end for**
10:    **end for**
11:    **for** $k = 1, \cdots, S$ **do**
12:       Obtain the representations (*e.g.*, $\hat{\mathbf{h}}_k$) of the generated outliers using our outlier generation method using Eq. (2)
13:    **end for**
14:    Compute the normal nodes' affinity $\tau(\mathcal{V}_o)$ and the outlier nodes' affinity $\tau(\mathcal{V}_l)$
15:    Compute the structural affinity loss $\ell_{ala}$ and egocentric closeness loss $\ell_{ec}$ using Eq. (4) and Eq. (5) respectively.
16:    Compute the BCE loss function $\ell_{bce}$ for our one-class classifier $\eta(\mathbf{h}_i; \Theta_s)$ using Eq. (6)
17:    Compute the total loss $\ell_{total} = \ell_{bce} + \beta\ell_{ala} + \lambda\ell_{ec}$
18:    Update the weight parameters $\mathbf{\Theta}$, $\Theta_g$ and $\Theta_s$ by using gradient descent
19: **end for**
20: **for** each $v_i$ in $\mathcal{V}_{test}$ **do**
21:    Anomaly scoring by $\mathbf{s}(v_i) = 1 - \eta(\mathbf{h}_j; \Theta^*)$
22: **end for**
23: **return** Anomaly scores $\mathbf{s}(v_1), \cdots, \mathbf{s}(v_{N_u})$

---

---

**Algorithm 2** Mini-Batch Processing

---

**Input**: Graph, $\mathcal{G} = (\mathcal{V}, \mathcal{E}, \mathbf{X})$, $N$: Number of nodes, $t$: Batch size, $z$: Number of batches, $S$: The number of the generated outlier nodes, $\mathcal{V}_{train}$:Training set
   **Output**: Mini-batches and sub-graph structure.

1: Initialize the batch $\mathbf{B} = (\mathbf{b}_1, ..., \mathbf{b}_z)$, where $\mathbf{b} = [v_1, ..., v_t]$ from given $\mathcal{V}_{train}$
2: **for** each $\mathbf{b}$ in $\mathbf{B}$ **do**
3:    Sample $S/z$ nodes as initial outliers $[v_{o_1}, ... v_{o_S}]$ from batch $\mathbf{b}$
4:    Initialize a node set $\mathcal{V}_\mathbf{b}$ for batch $\mathbf{b}$
5:    **for** each $v$ in $\mathbf{b}$ **do**
6:       Find the 2-hop neighborhoods $\mathcal{N}_v^2$ of $v$ and add them into the node set $\mathcal{V}_\mathbf{b}$
7:    **end for**
8:    Build a sub-graph structure $\mathcal{E}_\mathbf{b}$ for batch $\mathbf{b}$ using the node set $\mathcal{V}_\mathbf{b}$
9: **end for**
10: **return** The batch $\mathbf{B} = (\mathbf{b}_1, ..., \mathbf{b}_z)$ and sub-graph structure $[\mathcal{E}_1, ..., \mathcal{E}_z]$

---

