# OpenReview forum: "Generative Semi-supervised Graph Anomaly Detection"
_NeurIPS.cc/2024/Conference — NeurIPS 2024 poster_

### Official Review · Reviewer_7mdh · 2024-07-04

**Soundness:** 3
**Presentation:** 3
**Contribution:** 3
**Rating:** 5
**Confidence:** 4

**Summary:**

This paper works on node anomaly detection in the novel semi-supervised setting where few labeled normal nodes are given and proposes to generate new anomaly nodes to boost the training data. The anomaly generation algorithm is inspired by the empirical observation that:

(1) Anomaly nodes have lower affinity score than normal nodes
(2) Feature distribution of anomaly nodes are similar to normal nodes if they share similar neighborhood patterns.

**Strengths:**

(1) The setting is novel and aligned to the real-world situation where normal nodes are typically known compared with anomaly nodes.

(2) The motivation for the proposed two regularization losses is very intuitive and clear.

(3) The experimental results are very impressive.

**Weaknesses:**

(1) The proposed two regularization losses are heavily based on the empirical analysis, which might not transfer to other anomalies in other datasets.

(2) For the second prior, its assumption that anomaly nodes sharing similar local structures would share a similar feature distribution has not been empirically verified.

(3) Experiments miss the comparison with diffusion-based generative anomaly detection baseline.

**Questions:**

(1) As stated in the weakness, the core regularization loss terms are designed based on two assumptions:
* The anomaly nodes have a lower affinity score than normal nodes. However, there is no comprehensive experimental verification of the other datasets on this. It might be better to provide the verification like Figure 1 but on more different datasets.
* Anomaly nodes sharing similar neighborhood structures should possess similar feature distributions to their corresponding normal nodes. Although some references have been attached to justify this hypothesis, it might be better to include some empirical verification on this as well.

Furthermore, there might be some contradiction between these two assumptions by themselves. First, if assumption 1 holds, it means anomaly nodes should share different local subgraphs with the normal nodes, which indicates that assumption 2 cannot hold. How do we mediate this situation?

(2) Is there any difficulty when optimizing the loss according to Eq. (4) and Eq. (5) at the same time? Firstly, for Eq. (4), since the fixed terms would be embeddings of normal nodes and their neighbors, the embeddings of abnormal nodes ($\hat{\mathbf{h}}_i$ in Eq. (2)) would be optimized towards being further away from the neighbors' embeddings. However, Eq. (5) would also enforce the $\hat{\mathbf{h}}_i$ to be close to the normal one $\mathbf{h}_i$. These two directions seem to be contradictory to each other.

(3) Joint optimization according to Eq. (7) does not make sense under this generative augmentation setting. Here we use a generative model to augment the training data. This therefore should be that the training model is fixed. Moreover, if we jointly optimize the anomaly detection term and the other two generative terms, it would lead to the gradient for anomaly detection leaks to classification. This is quite confusing to me and might need more clarification.

(4) How many layers of the subgraphs are used in optimizing the affinity score? If we use 2-hop neighbors, it might cause the computation to consider the significantly large number of nodes. If not, how should we decide on this parameter?

(5) The comparison misses the baseline [1]

[1] Liu, Kay, et al. "Graph diffusion models for anomaly detection." (2024).

**Limitations:**

In addition to the limitations mentioned by the author, there are some other limitations worth addressing:

(1) The currently proposed anomaly generation method is still operated in the embedding space. As admitted by the author anomaly behavior is heavily based on interactional behaviors, therefore, it is also helpful to consider directly characterizing/generating anomaly in the graph space.

(2) The comparison misses one generative-based baseline [1]

[1] Liu, Kay, et al. "Graph diffusion models for anomaly detection." (2024).

---

> ### Author Rebuttal · Authors · 2024-08-07
>
> Thank you very much for the constructive comments and questions. We are grateful for the positive comments on the novelty and soundness of the experiments. Please see our detailed one-by-one responses below.
>
> > **Weaknesses #1** The regularization is heavily based on the empirical analysis, which might not transfer to other anomalies.
>
> The graph data is non-i.i.d. data with rich structure information, and the representation of a node should be grounded in a local context. Thus, as shown in Fig.1 and Fig.2 in the uploaded **pdf**,  these two important priors about graph anomalies generally hold in all these real-world graph anomaly detection datasets. This is also one main reason why the popular graph reconstruction and graph contrastive learning methods for GAD are generally effective on various GAD datasets, since their intuitions are essentially based on similar priors as ours (though they did not explicitly summarize and propose the priors).
>
> We agree that there can be some anomalies that may not well conform to these two priors. GGAD can work well for these cases too. This is because the outlier nodes generated by GGAD are essentially located at the fringe of the normal nodes in the representation space, as shown in Fig.3 (c). GGAD then leverages these outlier nodes to build a one-class classifier with a decision boundary tightly circled around the normal nodes. Such decision boundary can discriminate not only the anomalies well simulated by the outlier nodes but also the other anomalies that lie at the same side of the outlier nodes.
>
> > **Weaknesses #2** and **Questions #1**  More verification on more datasets and empirical verification of the second prior
>
> Please refer to our reply to **Global Response to Share Concern #1** in the overall author Rebuttal section above for this concern.
>
>
> > **Weaknesses #3** and  **Limitations #2** and **Questions #5** The lack of diffusion model based generation
>
> Thanks for pointing out the related study, but the work and other diffusion-based GAD methods as well focus on the fully supervised setting where both labeled normal and abnormal nodes are required during the training. This is different from the semi-supervised setting we proposed in our paper. We will discuss and include this work in our revision.  Additionally, we have added more ablation studies in the outlier generation module, please refer to the response to Reviewer JEU8's Question #4.
>
>
> > **Questions #2**  How do we mediate the two assumptions that might be some contradiction and is there any difficulty in optimization?
>
> Please refer to our reply to **Global Response to Share Concern #2** in the overall author rebuttal section above for this concern.
>
> > **Questions #3**  Clarification on the joint optimization with anomaly detector
>
> Different from existing augmentation-based generation, since we generate the outliers in the latent space rather than the raw node attribute space, the joint optimization allows the model to impose both anomaly priors more effectively.  The total loss generally decreases and converges finally,  see Fig. 3 in the uploaded **pdf**. To further investigate the benefits of joint optimization, we compare the two-step optimization method with our joint optimization approach. The experimental results are shown in Table A1. The results show that the joint optimization significantly outperforms the two-step approach. The main reason is that many more effective outlier samples can be generated during the optimization process of the two anomaly priors. By jointly optimizing with the BCE loss function, these outliers can be fully exploited for the training of the one-class classifier, enabling a mutually enhanced outlier node generation and one-class classification process.
>
> ```
> Table A1. Results of two-step training and joint training  (AUPRC/AUROC).
> ```
> |**Data**|**Amazon**|**T-Finance**|**Reddit**|**Elliptic**|**Photo**| **DGraph**|
> |:----   |:----:    |:----:       |:----:     |:----:     |:----:    |:----:    |
> | Two-step   | 0.1899 /0.7322   | 0.0805/0.7022     | 0.0399/0.5233     |  0.2282/0.6873   |0.1179/0.6089      |0.0050/0.5392     |
> | GGAD(Joint)|**0.7922/0.9443** | **0.1825/0.8228** | **0.0610/0.6354** | **0.2425/0.7290**| **0.1442/0.6476** |**0.0082/0.5943** |
>
>
> > **Questions #4** The number of layers of the subgraphs, how should we decide on this parameter
>
> We used one layer subgraph by default as the egonet of the target node to calculate the affinity score, because anomalies are primarily reflected in the relationships among the neighbors immediately connected to themselves [Ref1-2].  We agree that incorporating 2-hop neighbors can include more community information for GAD but might cause more computational overhead. We leave it as a promising future work direction.
>
>
>
> > **Limitations #1** The generation is still operated in the embedding space, heavily based on interactional behavior.
>
> Thank you very much for pointing out this.  Generation in the latent space is a simple yet effective way to support efficient generative GAD training. It can also mitigate the notorious over-smoothing representation problem in the GNN aggregation.
>
> Generating in the graph space is also a promising direction that offers a valuable approach for analyzing characteristics and enhancing interpretability. We will include and discuss this limitation in the final paper.
>
> **References**:
>
> - [Ref1] Addressing Heterophily in Graph Anomaly Detection: A Perspective of Graph Spectrum, WWW2023
> - [Ref2] Truncated Affinity Maximization: One-class Homophily Modeling for Graph Anomaly Detection, NIPS2023

---

> > ### Comment · Reviewer_7mdh · 2024-08-11
> > **Appreciate your response!**
> >
> > Thank you for more analysis. I have follow-up questions as follows:
> >
> > **Weaknesses #2 and Questions #1 More verification on more datasets and empirical verification of the second prior**: after seeing more visualization of normal/abnormal/gaussian-generated outliers distributions, I have the following questions: why would adding Gaussian noise make the outlier distribution like a single bar?
> >
> >
> > **Weaknesses #3 and Limitations #2 and Questions #5**: thank you for the clarification and it is reasonable.
> >
> >
> > **Questions #2 How do we mediate the two assumptions that might be some contradiction and is there any difficulty in optimization?**: I am not so convinced about the reply. Still, my initial concern is
> > - (1) One optimization is to make the generated node embedding itself similar to the original node embedding.
> > - (2) The other optimization is to make the generated node embedding itself have a lower affinity score than the neighbors of the original nodes
> >
> >
> > Let's assume (1) works and the generated node embeddings become further away from neighbor embeddings, then how would (2) still hold given that the original node embeddings have a higher affinity score to the neighbors?
> >
> >
> > Furthermore, I want to add on top of my above questions with another question given that this designed method needs to select normal nodes to generate abnormal nodes. How do we select the initial normal nodes? Do we use all of them or randomly sample them?
> >
> > Without further addressing my above concern, I cannot further raise my score.

---

> ### Author Response · Authors · 2024-08-12
> **Response to Reviewer 7mdh**
>
> We greatly appreciate your further comments and it's great to know that our response has helped address some of your questions. Please find our point-by-point response to your follow-up questions as follows.
>
>
> >**Questions #1** Why would adding Gaussian noise make the outlier distribution like a single bar?
>
> We'd like to clarify that the Gaussian noise works like a hyperparameter in the feature interpolation in Eq. (5) to diversify the outlier nodes in the feature representation space. Changes of this noise distribution do not affect the superiority of the detection performance of GGAD over the competing methods (please see the results in Table A1 in our response to Reviewer JEU8 for empirical justification). Further, the outliers are generated
> based on the neighbors of some sampled normal nodes, so the number of the outlier nodes is far less than that of all normal samples, leading to a relatively sparse distribution of the outlier nodes in the affinity density map compared to the normal nodes in the figure. However, this does not affect their effectiveness in serving as negative samples for training the one-class classifier.
>
>
> >**Questions #2** Concern on two constraints
>
> We achieve the mediation of two constraints by jointly minimizing $L_{ala} $ and $L_{ec}$. The mediation leads to a result where the generated outlier nodes lie at the fringe of the normal nodes in the feature representation space, as illustrated in Fig. 1(b) and Fig. 3(c). Such outlier nodes meet the two criteria evenly you mentioned in the above comment.
> In terms of optimization, from Fig.3 in the uploaded **pdf**, we can see that the overall loss and the two individual losses gradually converge at around similar epochs where both constants are satisfied for the generated outliers; not unstable or fluctuated bounces are found after these epochs. Thus, we didn't experience any difficulty in the optimization of GGAD across all the datasets used.
>
>
> >**Questions #3**  How do we select the initial normal nodes?
>
> As mentioned in lines 202-204, We randomly sample a set of *S* normal nodes and respectively generate an outlier node for each of them based on its ego network. In line 300, we indicate the size of the generated outlier node S is set to 5\% of by default.  As shown in Fig. 6 and Fig. 7 in App C.1 of the paper, the performance of GGAD generally remains stable w.r.t the number of the generated outlier nodes.
>
> We hope the above replies help address your concerns. We're more than happy to engage in more discussion with you to address any further concerns you may have.
>
> Thank you very much for helping enhance our paper again!

---

> ### Comment · Reviewer_7mdh · 2024-08-12
> **Follow-up further**
>
> **Question #1: Why would adding Gaussian noise make the outlier distribution like a single bar?**
>
> My question here is not whether tweaking the Gaussian noise would enhance previous baseline performance but more like if we can tweak the added Gaussian noise, would the motivation figure also suffer from this issue (the distribution of generated outliers in previous methods are further away than ground truth ones compared to the proposed methods)? However, since the presented results show significantly better performance of the proposed method, I guess it should be that no matter what level of Gaussian noise we add, they would still not be more like the ones generated by the proposed method.
>
> **Question #3: How do we select the initial normal nodes?** This makes sense to me. Thanks!
>
> **Question #2 Concern with two constraints** I am still confused here and would hear more guidance on this point. Because neighbors embeddings and original node embeddings are fixed and assuming we are working on the homophily social networks (which is a widely adopted property in real-world datasets), how could we enforce the generated outlier node embeddings to be further away from neighbors but at the same time be close to the original center node.
>
> Although I still have questions on **Question #2 Concern with two constraints**. Overall, I appreciate the authors' response and think the observation of this paper would still be useful in future research in anomaly detection. I increase my score but hope authors could further address my questions here **Question #2 Concern with two constraints**.

---

> > ### Author Response · Authors · 2024-08-13
> > **Response to Reviewer 7mdh**
> >
> > Thank you very much for raising the score. We greatly appreciate your further comments and it's great to know that our response has addressed most of your questions. Please find our response to this concern.
> >
> > >**Follow-up question with Question #1** Why would adding Gaussian noise make the outlier distribution like a single bar?
> >
> > Yes, your understanding is correct. Changing the level of Gaussian noise in the generation of the outlier nodes in the baseline methods does not affect the detection performance of the baseline methods AEGIS and GAAN, since they still suffer from the absence of considering graph structure information in their generation model.
> >
> > >**Follow-up question with Question #2** Concern with two constraints
> >
> > We agree that in the homophily graph, the nodes tend to connect with the nodes from the same class, leading to the relatively high similarity between the embeddings of the target node and their neighbors. However, this does not affect the learning of the outlier nodes our method aims to obtain. This is because, given the embeddings of a fixed target node and their neighbors, minimizing the two proposed losses jointly would result in a mediation in the feature representation space where the generated outlier nodes are close to, yet separable from, the target normal node and its neighbors. Thus, these outlier nodes can be thought as `hard anomalies` that lie at the fringe of normal nodes in the feature representation space. If the egocentric closeness loss is removed, the generated outliers would become `trivial anomalies` that distribute far away from the normal nodes (see Fig. 3(a) in the paper). On the other hand, if the local affinity loss is removed, the generated outliers would then become `misleading/false anomalies` that lie inside the normal nodes in the feature representation space (see Fig. 3(b) in the paper).
> >
> > We hope the above reply helps address your follow-up questions. We will clarify this point in our final version. We're more than happy to engage in more discussion with you to address any further questions you may have. Thank you very much for helping enhance our paper again!

---

### Official Review · Reviewer_R8Ff · 2024-07-09

**Soundness:** 3
**Presentation:** 3
**Contribution:** 3
**Rating:** 6
**Confidence:** 3

**Summary:**

The paper proposes a novel approach called GGAD aimed at improving anomaly detection in graphs under a semi-supervised framework. GGAD generates pseudo anomaly nodes that serve as negative samples for training a one-class classifier. This method is built on two
key priors: asymmetric local affinity and egocentric closeness, which help in generating reliable outlier nodes that mimic real anomalies in terms of both graph structure and feature representation. Extensive experimental results demonstrate the effectiveness of the method across diverse graph anomaly detection datasets.

**Strengths:**

1.The method is innovative. The proposed graph anomaly detection method can exploit the feature and structure information of normal nodes more effectively in the studied semi-supervised scenario compared to existing methods.  The proposed two priors provide a meaningful characterization of desired properties of outliers in this semi-supervised setting and can be utilized to explore other beneficial priors further.

2.The experiments in the paper are comprehensive and thorough.

**Weaknesses:**

1. The model relies on prior knowledge to generate anomaly points. This prior knowledge can limit the model’s application scenarios. The model performs best only when the real anomalies align with this prior knowledge. For anomaly types that do not conform to the prior knowledge, the model may not effectively detect them.

2.The model does not perform best on the Photo dataset in Table 1, and the article lacks an explanation of the results at the overall data level.

3. This model employs a semi-supervised approach that uses some positive samples for training. However, it does not consider the issue of noise interference within the positive samples, namely, how the model overcomes interference when some positive samples are mislabeled.

4. During the initialization step, only the initial feature of outliers are obtained while the connections between the outliers and normal nodes are not well illustrated in the paper. From Figure 2, one outlier is connected to more than one normal node while the feature of the outlier is generated according to single normal node. The neighborhood of outliers is important since the it involves the computation of node affinity score of outliers.

**Questions:**

see weakness

**Limitations:**

yes, the authors point out that some anomalies whose characteristics may not be captured by the two priors used

---

> ### Author Rebuttal · Authors · 2024-08-07
>
> Thank you very much for the constructive comments. We are grateful for the positive comments on our paper clarity, research motivation, and empirical justification. Please see our response to your comments one-by-one below.
>
> > **Questions #1** The anomalies do not conform to the prior knowledge
>
> Please also refer to the response to Review 7mdh Question #1 for detailed clarification.
>
>
> > **Questions #2** Analysis of the results of the Photo and the results at the overall data level
>
> Thank you very much for the comment and suggestion. GGAD yields the best AUROC on the Photo while yielding the second-best in AUPRC, underperforming OCGNN. Having the best AUROC while less effective AUPRC indicates that GGAD can detect some anomalies very accurately in Photo, but it is less effective than OCGNN to get a bit more anomalies rank in the top of normal nodes in terms of their anomaly score.
> We will add the above discussion into the final paper.
>
> > **Questions #3** How the model overcome mislabeled positive samples
>
> To consider this issue, we introduce a certain ratio of anomaly contamination into the
> training normal node set to simulate the normal nodes that are mislabeled in our experiments.  The results of the models under different ratios of contamination in Fig. 5. and App. C.3 shows that with increasing anomaly contamination, the performance of all methods decreases. Despite the decreased performance, our method GGAD consistently maintains the best performance under different contamination rates, showing good robustness w.r.t. the contamination/noise. The main reason is that, unlike most unsupervised methods, GGAD not only learns normal patterns through normal models but also learns abnormalities by generating anomalies based on two priors, which reduces the dependency on the quality of normal data.
>
> > **Questions #4**  The connection between the outliers and the normal node is not well-illustrated
>
> In the initialization step, we first sample some normal nodes and generate the outliers based on the representations of the neighbors of each target normal node, so the generated outlier nodes share similar neighborhood information as the target normal nodes. We will add more detailed explanations for Fig.2 in the final paper to clarify this point.

---

### Official Review · Reviewer_JEU8 · 2024-07-10

**Soundness:** 2
**Presentation:** 3
**Contribution:** 2
**Rating:** 5
**Confidence:** 5

**Summary:**

This paper introduces a novel generative-based GAD approach, named GGAD, tailored for the semi-supervised scenario. Unlike existing GAD frameworks, the authors highlight the feasibility and importance of a semi-supervised setting where labels for normal nodes are relatively easy to obtain during training, but labeled abnormal nodes are very limited. In this context, the paper proposes generating pseudo-anomaly nodes to serve as substitutes for real anomaly nodes in training, thus aiding in anomaly detection. These pseudo-anomalies are generated through two unique loss-guidance mechanisms. Experimental results demonstrate the effectiveness of GGAD.

However, the description of the semi-supervised setting in this paper lacks clarity and unconvincing. Additionally, there is minimal differentiation between the proposed method and existing works that generate pseudo-anomaly samples for data augmentation. I think this paper's novelty is limited. I still think that doing unsupervised GAD is more necessary, and if the authors can prove that the pseudo-outlier proposed by GGAD can benefit unsupervised GAD as a general module, I can up my score.

**Strengths:**

1.The complete experiment shows the effectiveness of the method and the necessity of each component.

2.Some visual illustrations help the reader understand, although the shapes of the images seem to be compressed.

**Weaknesses:**

1. I am still confused about the motivation for performing semi-supervised GAD. Why do most methods emphasize unsupervised scenarios? The cost of labeling normal nodes seems too expensive, as the authors themselves state on lines 268 to 269, yet they assert again on line 31 that labels for normal nodes are easy to obtain.This inconsistency hinders a clear understanding of the necessity and practical applications of semi-supervised GAD, which significantly undermines the motivation for this work.

2. While the first loss function proposed by the authors appears intuitively valid, the second loss function aims to generate outliers similar to normal nodes. In my opinion, optimizing these two losses together is unreasonable because they conflict with each other. It seems that they should correspond to different outlier generation processes

3. The paper validates the improvement of unsupervised GAD using labeled normal nodes and claims that GGAD remains superior. I think the authors ignore the fact that unsupervised methods do not obtain this outlier like GGAD and this comparison is not reasonable.

**Questions:**

1. why semi-supervised GAD is more important than unsupervised GAD, How do you overcome the labeling cost?
2. If unsupervised GAD methods use outliers in GGAD, is it beneficial for them?
3. why Eq.5 need Gaussian noise?
4.In addition to the outlier generation methods mentioned on lines 376-396 (they seem overly simplistic), are there more advanced methods for generating outliers similar to GGAD? How does GGAD compare to them?

**Limitations:**

No limitation need to discuss

---

> ### Author Rebuttal · Authors · 2024-08-07
>
> Thank you very much for the constructive suggestions.  We are grateful for the positive comments on our readability and empirical justification. Please see our response to your comments one by one below.
>
>
> > **Weakness in Summary** Minimal differentiation with existing generation pseudo-anomaly samples (Benefits to unsupervised GAD).
>
> Our work is the first method that leverages the priors related to graph anomalies into the outlier node generation, which enables the generation of outlier nodes considering both graph structure and feature representations of real abnormal nodes, which is not viable to any existing outlier/pseudo-anomaly generation methods (We will show how GGAD can benefit unsupervised GAD as an outlier generation module in our reply to Question #2 below).
>
>
> > **Weaknesses #1** and **Questions #1** The motivation of Semi-supervised GAD and labeling cost
>
> Thank you very much for the comment. Most of the methods are based on unsupervised scenarios since it requires no labeling cost. However, they are too restricted in the real applications, because labels for a small set of normal nodes are easy to obtain. This is mainly due to the fact that the number of normal nodes typically overwhelmingly dominates the full graph. Thus, for example, one can randomly sample some nodes from the graph as the normal nodes, without any human manual labeling (the same labeling cost as unsupervised GAD). The quality of this 'random'  normal labeling is high considering the scarcity of anomalies. Even involving human checking, such a small node set does not require much effort.
>
> The randomly labeled normal nodes may include a very small number of abnormal samples in some cases. This is also why we perform extensive experiments on such cases in Fig. 6, where the detectors are evaluated on anomaly-contaminated normal data.
>
> In line 31, we stated that ``the normal nodes are easier to obtain due to their overwhelming presence in the graph``. We will rephrase that ``a small part of normal nodes is easier to obtain`` to avoid misunderstanding. In lines 268-269 we meant that human checking of large-scale normal nodes can be costly, contrasting to that for a small part of the normal nodes.
>
>
>
> > **Weaknesses #2**  Optimize these two losses seems to conflict and they should correspond to different outliers
>
> Please refer to our reply to **Global Response to Share Concern #2** in the overall author Rebuttal section.
>
>
> > **Weaknesses #3** and **Questions #2**  Ignore the fact that unsupervised GAD methods do not obtain the outlier like GGAD and if GGAD can benefit existing unsupervised methods as a general module
>
> Incorporating the outlier generation into existing unsupervised methods would lead to fairer empirical comparison.  To allow the unsupervised methods to exploit the generated outliers, we first utilize GGAD to generate outlier nodes by training on randomly sampled nodes from a graph (which can be roughly treated as all normal nodes due to anomaly scarcity) and then remove possible abnormal nodes from the graph dataset by filtering out Top-K most similar nodes to the generated outlier nodes.  By removing these suspicious abnormal nodes, the unsupervised method is expected to train on the cleaner graph (i.e., with less anomaly contamination).
> This approach to improve unsupervised GAD methods is referred to as GGAD-enabled unsupervised GAD. We evaluate their effectiveness on three large-scale datasets.  Please refer to **global response** for the results in Table A1, where  #Anomalies/#Top-K Node in the table respectively represents the number of real abnormal nodes we successfully filter out and the number of nodes we choose to filter out (i.e., K).
>
> For example, we use the outlier nodes generated by GGAD to filter out 500 nodes from the Amazon dataset, of which there are 387 real abnormal nodes. This helps largely reduce the anomaly contamination rate in the graph. The results show that this approach can significantly improve the performance of three different representative unsupervised GAD methods, including DOMINATE, OCGNN, and AEGIS.
>
> Note that although the GGAD-enabled unsupervised methods achieve better performance, their performance still largely underperforms GGAD, which provides stronger evidence for the effectiveness of GGAD.
>
> > **Qestions #3** The role of Gaussian noise in Eq.5
>
> This simple perturbation can help maintain the affinity separability while enforcing the egocentric closeness constraint.
>
> > **Questions #4** Compare with more advanced methods for generating outliers similar to GGAD
>
> Apart from the outlier generation in the ablation study, we further employ two advanced generation approaches, VAE and GAN.  In VAE, we generate the outlier representations by reconstructing the raw attributes of selected nodes where our two anomaly prior-based constraints are applied to the generation. For GAN. we generate the embedding from the noise and add an adversarial function to discriminate whether the generated node is fake or real, with our two prior constraints applied in the generation as well. As shown in Tab.1 in uploaded **pdf**. These two advanced generation approaches can work well on some datasets, which indicates two priors help them learn relevant outlier representations. But both of them are still much lower than GGAD, showcasing that the outlier generation approach in GGAD can leverage the two proposed priors to generate better outlier nodes.

---

> > ### Comment · Reviewer_JEU8 · 2024-08-11
> >
> > Dear Authors:
> >
> > On the semi-supervised question, I assume that you only use a subset of nodes in the original dataset as normal samples and use the generated abnormal samples (similar to data augmentation?) to train a classifier, if that's the case, I think it's reasonable and would be willing to boost my score to 4.
> >
> > However, I think the process of generating abnormal nodes seems too simple, I can understand the intention of Eq(4), but how to use abnormal nodes to calculate Eq(3), how do you determine what their neighbors have? I think that's critical. Eq (5) is more like a regularization term and I think its contribution is small.

---

> > > ### Author Response · Authors · 2024-08-11
> > > **Response to Reviewer JEU8**
> > >
> > > We greatly appreciate your further comments and it's great to know that our response has helped address your questions. Please find our point-by-point replies as follows.
> > >
> > > **(0)** Setting.
> > >
> > > Yes, the studied setting assumes the availability of a small set of labeled normal nodes during training. To train a discriminative one-class classifier, our method GGAD utilizes these normal nodes to generate pseudo abnormal nodes that assimilate real anomaly nodes in both graph structure and feature representations.
> > >
> > > **(1)** Similar to data augmentation?
> > >
> > > Current data augmentation methods for GAD like DAGAD and DIFAD [Ref1-2] and imbalanced graph learning like GraphSMOTE and GraphMixup [Ref3-4] require both labeled abnormal nodes and normal nodes, making them not applicable to the studied semi-supervised setting where only the labels of partial normal nodes are available during training. Unlike these graph data augmentation methods, our GGAD is a generative method that generates pseudo abnormal nodes by leveraging the two abnormality-related priors using only a small set of normal nodes. Furthermore,  from the ablation study results in Table 3 in the paper, commonly used graph data augmentation methods like the random sampling or mixing up normal nodes with noise cannot work. Besides, the newly added results of existing popular generative methods in Table 1 in the upload **pdf** further demonstrate the unique effectiveness of the proposed generative method in our GGAD.
> > >
> > >
> > >
> > > **(2)** The calculation of Eq(3) for abnormal nodes ?
> > >
> > > As mentioned in lines 202-204, and shown in Fig. 1 left in the paper, we sample some normal nodes as the target location for outlier node generation, in which the outlier nodes share the same neighbors with the sampled target normal nodes. The affinity calculation of the generated abnormal nodes is based on the neighbors of these target normal nodes.
> > >
> > > **(3)** Eq. (5) is more like a regularization term and I think its contribution is small.
> > >
> > > Eq. (5) is designed to incorporate our egocentric closeness-based abnormality prior, through which we aim to pull the representations of generated outliers close to the node representations in its egonet in the feature representation space, as shown in Fig. 3(c). Without this loss, the generated abnormal nodes only meet one abnormality prior, which do not provide sufficient discriminative information for the one-class classifier. Thus, the contribution of this loss is significant: it is not only reflected in its own design but also in the collaboration with the other prior implemented via Eq. (4). This collaborative effect helps generate the pseudo abnormal nodes which can serve as effective negative samples for training a tight decision boundary of the one-class classifier.
> > >
> > > As the very first work to explore the semi-supervised GAD setting, we introduce a simple yet effective way to generate pseudo abnormal nodes for training an accurate one-class classifier for GAD, offering a novel principled framework for semi-supervised GAD. Further, our method GGAD can also perform well in anomaly-contaminated training data; as also pointed out by you and justified by our newly added results, the generated abnormal nodes can also help improve unsupervised GAD methods in the popular unsupervised setting. All these lead to a piece of work that has significant contributions to GAD in the newly introduced semi-supervised setting and the widely-explored unsupervised setting. In terms of methodology, all modules in our method are novel for GAD, and as far as we know, GGAD is the very first method that can generate pseudo abnormal nodes assimilating real anomaly nodes in both graph structure and feature representations. It is true that GGAD is simple, but as per Occam's razor, simpler models are preferred over more complex ones. Thus, we argue that our method is technically solid and expected to have moderate-to-high impact in the GAD community.
> > >
> > > We're more than happy to engage in more discussion with you to address any further concerns you may have.
> > >
> > > Thank you very much for helping enhance our paper!
> > >
> > > **References**:
> > > - [Ref1] DAGAD: Data Augmentation for Graph Anomaly Detection， ICDM2022
> > > - [Ref2] NEW RECIPES FOR GRAPH ANOMALY DETECTION: FORWARD DIFFUSION DYNAMICS AND GRAPH GENERATION, 2024
> > > - [Ref3] G-Mixup: Graph Data Augmentation for Graph Classification ICML2022
> > > - [Ref4] GraphSMOTE: Imbalanced Node Classification on Graphs with Graph Neural Networks WSDM2021

---

> > > > ### Comment · Reviewer_JEU8 · 2024-08-11
> > > >
> > > > Dear Authors:
> > > >
> > > > Thank you for your reply. I am glad that my understanding of problem setting is consistent with yours. If you only use a small number of normal nodes as the training set, I think it is reasonable and i will raise my score.
> > > >
> > > > I acknowledge Occam's razor and the fact that it is critical to an ML-based model, especially one based on neural networks. However, I still have some concerns about the current neighbor selection, and the fact that many anomaly detection methods are based on the structure of the graph, which you ignore, may affect the scope of the method (it cannot be used on graphs without attributes).Also, regarding Gaussian noise, I think there is a lack of a mathematically explanation, which further adds to my concerns about the method.
> > > >
> > > > To sum up, I think anomaly detection is an important area, so I would like to critically examine the reasonability and feasibility of related methods, for which I do not support accepting this paper, I will keep my score: 4

---

> ### Author Response · Authors · 2024-08-12
> **Response to Reviewer JEU8**
>
> Thank you for raising the score. Please find our point-by-point replies as follows.
>
> > **Questions #1** Ignore the fact that many anomaly detection methods are based on the structure of the graph, which may affect the scope of the method (it cannot be used on graphs without attributes)
>
> As mentioned above, our GGAD generates outlier nodes that assimilate the real anomaly nodes in both graph structure and feature representations, where the structure information of the graph has been fully considered in our methodology design (i.e., through Eqs. (2) and (3) in the asymmetric local structural affinity prior). As emphasized in many GAD studies, anomalies in the graph are primarily reflected in the relationships of a node to its neighboring nodes immediately connected to themselves [Ref1-2] and many other methods reviewed in [Ref3]. That's why we consider generating the outlier nodes based on the egonet of target normal nodes.
>
> Currently, to our best knowledge, most GAD methods are focused on attributed graph datasets (please see the survey and benchmark papers in [Ref3-5] for more details). As for the graph without attributes, the methods for attributed graphs can be applied by augmenting the graph with node attributes based on, e.g., one-hot encoding based on neighborhood information or feature construction using graph structure information (see [Ref3]). Thus, this should not be seen as a limitation of our method and numerous existing GAD methods.
>
>
> > **Questions #2** The implementation of Gaussian noise
>
> This simple perturbation can help maintain the affinity separability while enforcing the egocentric closeness constraint. It is a hyperparameter in GGAD and the default value was presented in the implementation. Gaussian noise-based perturbation is commonly used in existing feature interpolation techniques, including those in GAD methods [Ref6-7], and it is used as a way to diversify the generated outlier nodes only, not having catastrophic impacts on the performance of GGAD. This is justified by the newly added results in Table A1 below where we change the mean and variance of the Gaussian noise and replace the Gaussian noise with uniform noise. The results show that, regardless of the distribution of the noise, GGAD remains very effective, demonstrating similar superiority over the competing methods in Table 1 in the paper.
>
> ```
> Table A1. The performance of GGAD under different scales of Gaussian noise and uniform noise (AUPRC/AUROC).
> ```
> |**Data**| **Amazon**|**Elliptic**|**Photo**|
> |:----   |:----:    |:----: |:----:     |
> | mean=0, std = 0          |     0.7343 / 0.9192     |     0.2107 / 0.7060    |   0.1325/0.6432  |
> | mean=0.005, std = 0.001    |     0.7475  /  0.9233   |    0.2240  /  0.7110   |   0.1401/0.6444  |
> | mean=0.01, std = 0.005   |     0.7834  / 0.9324    |    0.2425 / 0.7290     |   0.1442/0.6476  |
> |uniform noise(a=0, b=0.01)       |    0.7434 / 0.9142          |    0.2173  /0.7163     |   0.1407/0.6526   |
>
>
> Overall, in our paper and the rebuttal here, we have examined the feasibility of a very wide range of methods that are related to our method GGAD from diverse aspects, e.g., different ways of exploiting graph structure information, generating outlier nodes, implementing GGAD with various alternative methods, and utilizing the generated outlier nodes in unsupervised/semi-supervise settings, etc. Thus, we would greatly appreciate if you could provide other unexplored directions for us to further evaluate the effectiveness of our method GGAD. We would kindly request your reconsideration of your current rating of our paper if otherwise. Thank you very much again!
>
> **References**:
>
> - [Ref1] Addressing Heterophily in Graph Anomaly Detection: A Perspective of Graph Spectrum, WWW2023
> - [Ref2] Truncated Affinity Maximization: One-class Homophily Modeling for Graph Anomaly Detection, NIPS2023
> - [Ref3] A comprehensive survey on graph anomaly detection with deep learning, TKDE 2021
> - [Ref4] GADBench: Revisiting and Benchmarking Supervised Graph Anomaly Detection, NIPS2023
> - [Ref5] BOND: Benchmarking Unsupervised Outlier Node Detection on Static Attributed Graphs NIPS2022
> - [Ref6] Perturbation learning based anomaly detection, NIPS2022
> - [Ref7] DAGAD: Data Augmentation for Graph Anomaly Detection， ICDM2022
> - [Ref8] Consistency Training with Learnable Data Augmentation for Graph Anomaly Detection with Limited Supervision, ICLR2024

---

> > ### Comment · Reviewer_JEU8 · 2024-08-12
> >
> > Dear Authors:
> >
> > Thanks for your reply! As you said, for graphs without attributes, you can add hand-crafted features, which is normal and not a shortcoming of GGAD. My initial concern was that it seemed unreasonable to let the generated abnormal nodes share the same neighbors as the normal nodes. I read the paper again, and I thought that the loss in Eq 4 might solve this problem. I can accept the author's explanation.
> >
> > But with Gaussian noise, If you say it's "a way to diversify the generated outlier nodes only, not having catastrophic impacts on the performance of GGAD. "Then I still don't think it's the most necessary.
> >
> > To sum up, the loss in Eq 4 is good and dispels my confusion, but the loss in Eq 5 weakens the novelty, so I'm sorry that I can't improve the score.

---

> > > ### Author Response · Authors · 2024-08-12
> > >
> > > Dear  Reviewer JEU8,
> > >
> > > Thanks a lot for the prompt reply. We apologize that we don't understand why the use of the loss in Eq. 5 weakens the novelty of our method, given the fact that you find good novelty in Eq. 4.
> > >
> > > As demonstrated by our ablation study results in Table 2 in the paper, the adding of the loss in Eq. 5 to our method leads to very significant performance improvement across all the datasets in both AUROC and AUPRC. Moreover, as shown in the newly added results in Table A1 above, the loss in Eq. 5 works well regardless of different prior distribution used to specify the noise. We would really appreciate if you could kindly advise why the additional major contribution made by the loss in Eq. 5 is considered as a negative part of our model design. Thank you!

---

> > > > ### Comment · Reviewer_JEU8 · 2024-08-12
> > > >
> > > > Dear Authors:
> > > >
> > > > I'm sorry for the misunderstanding caused by my reply. I don't think Loss in Eq. 5 is considered as a negative part of our model design. I acknowledge that your experiments prove its effectiveness, I read the paper again, my opinion is, if you are aim to generate nodes that are less like anomaly nodes, more like normal nodes, then I think you need to select the failed nodes instead of computing on all nodes as in Eq. 5.

---

> > > > > ### Author Response · Authors · 2024-08-12
> > > > >
> > > > > Dear Reviewer JEU8,
> > > > >
> > > > > Thanks for the prompt clarification. However, we believe there are some major misunderstanding here.
> > > > >
> > > > > - First of all, our goal is always to generate outlier nodes assimilating real anomaly nodes, but we focus more on generating the `hard' anomaly nodes that lie at the fringe of the normal nodes in the feature representation space, corresponding to our egocentric closeness prior. This is why the loss in Eq. 5 plays a crucial role in our outlier generation. Conversely, if we remove the loss in Eq. 5 from our objective function, the generated outliers would be simply trivial outlier nodes that are far away from the normal nodes, which are not that helpful for learning a discriminative one-class classifier, as supported by our results in Fig. 3 and ablation study results in Table 2 in the paper.
> > > > >
> > > > > - Secondly, the number of normal nodes or the type of normal nodes we use in Eq. 5 does not really affect the performance. This is because the normal nodes in the feature representation space are generally closed to each other. Please also note that the normal nodes used in Eq. 5 are randomly sampled from the labeled normal node set. Increasing or decreasing the number of normal nodes has limited impact on the performance of GGAD, as shown in Fig. 6 and Fig. 7 in App C.1 of the paper.
> > > > >
> > > > > We hope these clarifications help sort out the possible misunderstanding you might have before. We apologize for the confusion and will clarify these points in our final paper. Thank you very much again!

---

> > > > > > ### Comment · Reviewer_JEU8 · 2024-08-12
> > > > > >
> > > > > > Dear Authors:
> > > > > >
> > > > > > I agree with your explanation this time, and I think you need to add the difference between "hard" outliers and "trivial" outliers to the manuscript, which is the key motivation of Eq 5 in my opinion. I will raise my score to 5, and I hope my suggestion will be helpful to improve the quality of the paper.

---

> ### Author Response · Authors · 2024-08-12
>
> Dear Reviewer JEU8,
>
> We're very pleased that our clarification is helpful, and thank you for increasing the rating to an acceptance score.
>
> We will add the intuition of "hard" outlier nodes and its difference to "trivial" outlier nodes in Sec. 3.4 to clarify why the loss in Eq. 5 is important in our method. The discussions have been very helpful for enhancing our paper. Many thanks again for your time and effort on our paper.  We're happy to take any further questions you might have.

---

### Official Review · Reviewer_yDqR · 2024-07-11

**Soundness:** 2
**Presentation:** 3
**Contribution:** 2
**Rating:** 5
**Confidence:** 3

**Summary:**

This paper explores the problem of semi-supervised graph anomaly detection (GAD), where some nodes are known to be normal, in contrast to the typical unsupervised setting with no labeled data. The authors show that even a small percentage of labeled normal nodes can improve the performance of existing unsupervised GAD methods when adapted to the semi-supervised scenario. The paper proposes a novel Generative GAD approach (GGAD) to better exploit normal nodes by generating pseudo anomaly nodes, called 'outlier nodes', to provide effective negative samples for training a one-class classifier. GGAD generates these outlier nodes using priors about anomaly nodes, such as asymmetric local affinity and egocentric closeness, to mimic anomalies in structure and features. Experiments on six real-world GAD datasets show that GGAD outperforms state-of-the-art methods in both unsupervised and semi-supervised settings.

**Strengths:**

+ This paper studies a new problem of semi-supervised GAD that has not been widely studied.

+ The proposed method is simple and effective from the empirical perspective.

+ The experiments are extensive including effectiveness and efficiency analyses and the method has been tested on real-world large-scale graphs to verify the scalability.

**Weaknesses:**

- The two priors that are used to generate outlier nodes are heuristic or based on empirical evidence. There is no theoretical analysis provided to better guarantee the effectiveness of the proposed method.

- It will be more interesting and helpful to show the generated outlier nodes can capture the characteristics of anomalous nodes in addition to comparing their representations.

- The experimental settings of anomaly contamination are not very clear: how the contamination is introduced?

- Overall experimental settings. What hardware has been used in the experiments, e.g., memory, and why are the experiments conducted on CPUs?

**Questions:**

1. Theoretical analysis of the proposed method, especially these two priors.

2. Experimental settings including hardware and anomaly contamination.

3. Analysis of the generated outlier nodes.

**Limitations:**

The authors have adequately addressed the limitations

---

> ### Author Rebuttal · Authors · 2024-08-07
>
> Thank you very much for the constructive comments. We are grateful for the positive comments on our studied problem, technical contribution, and empirical justification. Please see our detailed response below
>
> > **Weaknesses #1**  There is no theoretical analysis to guarantee the effectiveness of the proposed method
>
> We encapsulate two important anomaly priors to generate the outliers that are similar in the local structure and feature representation as real abnormal nodes. This provides a principled framework for generative GAD. To further verify the intuition of the priors and our proposed method, we have added empirical evidence that justifies the two priors in more real-world GAD datasets in the uploaded **pdf**. Please refer to our reply to **Global Response to Share Concern #1** in the overall author rebuttal section above for details. Overall, our method, as the first piece of work explicitly designed for the semi-supervised GAD problem, presents solid findings and interesting insights into the problem, laying a good foundation for future work on theoretical analysis and more advanced methods in this research line.
>
>
> > **Weaknesses #2** and **Q3**  More analysis on the generated outliers like shot the generated outliers can capture the characteristic of anomalous in addition to comparing their representations
>
> Thank you very much for the suggestion. Please see Fig.3 in the paper and Fig. 1 in the uploaded **pdf** for the visualization based on the local structure information of the generated outlier nodes. We further employ the Maximum Mean Discrepancy (MMD) distance to measure the distance between the generated outliers and the real abnormal nodes (and the normal data as well) to illustrate more in-depth characteristics of the generated outlier nodes. As shown in Table A1 below, it is clear that the distribution of the generated outliers have much smaller MMD distance to the real abnormal nodes than the normal nodes, indicating the good alignment of the distribution of the generated outliers with the real abnormal nodes.
>
> ```
> Table A1. Analysis of the generated outlier nodes using MMD distance
> ```
> |**Data**|**Amazon**     |**T-Finance**   |**Elliptic** |  **Photo** |  **Reddit** |
> |:----   |:----:         |:----:          |  :----:   |  :----:   |:----:   |
> |with Abnormal Node|  **0.1980**  |  **0.0784**  |  **0.1094**   |   **0.3703**  |   **0.3409** |
> |with Normal Node  |    0.2318    |  0.1040      |  0.1304        |  0.3880       |  0.3605      |
>
> We will add this MMD distance-based outliers analysis into the final paper.
>
>
> > **Weaknesses #3** and **Questions #2** Clarification on the setting of contamination
>
> Thanks for pointing out the issue.  When studying the semi-supervised GAD, it's important to consider that some normal nodes may be mislabeled or affected by some noise interference. To introduce a certain ratio of anomaly contamination, we randomly sample $V_l$ nodes from the real abnormal nodes set in the dataset and add them as the contaminated nodes into the training data. The rest of the abnormal nodes are used as part of the test dataset. We will add these details of the contamination setting in the final paper.
>
> >**Weaknesses #4** and  **Questions #2**  Overall experimental settings. Provide more information about the hardware and why the experiments are conducted on the CPU.
>
> The existing baseline methods require different GPU memory and environments depending on their methodology design.  To conduct a unified comparison of operational efficiency, we chose an AMD EPYC 7443P 24-core CPU with 125G memory as the running platform. We also provided a computational efficiency analysis in Appendix D.

---

> > ### Author Response · Authors · 2024-08-12
> > **Kindly Request for Reviewer's Feedback**
> >
> > Dear Reviewer yDqR,
> >
> > Since the End of author/reviewer discussions is coming in ONE day, may we know if our response addresses your main concerns? If so, we kindly ask for your reconsideration of the score. Should you have any further advice on the paper and/or our rebuttal, please let us know and we will be more than happy to engage in more discussion and paper improvements.
> >
> > Thank you so much for devoting time to improving our paper!

---

> > ### Comment · Reviewer_yDqR · 2024-08-12
> > **Thanks for the rebuttal**
> >
> > I appreciate the efforts of the authors during the rebuttal phase. These responses addressed most of my concerns. But I still have some concerns for W4 and Q2: to make a fair comparison, why CPU is selected? Because of the large memory requirement to load a large graph?

---

> > > ### Author Response · Authors · 2024-08-13
> > > **Response to Reviewer yDqR**
> > >
> > > We're very pleased to know that our response has addressed most of your concerns. We really appreciate your further comments. Please find our response to your follow-up question as follows.
> > >
> > > >**Follow-up question with Question #1** To make a fair comparison, why CPU is selected? Because of the large memory requirement to load a large graph?
> > >
> > > Yes, your understanding is correct. A number of baselines like DOMINANT, AnomalyDAE, and TAM require large memory to handle large-scale graph datasets. However, we lack GPUs with a sufficiently large memory size to perform experiments on these large datasets using GPUs. Thus, in order to obtain the runtime results for all methods across all data sets using the same computing environment, we obtained the runtime results based on a consistent CPU-based setting.
> > >
> > > We hope the above reply helps address this follow-up question. We will clarify this point in our final version. We're more than happy to engage in more discussion with you to address any further concerns you may have. Thank you very much for helping enhance our paper again!

---

> > > > ### Comment · Reviewer_yDqR · 2024-08-13
> > > > **Thanks for the clarification**
> > > >
> > > > I appreciate the clarification from the authors. Now my concerns have been addressed and I would like to increase the rating.

---

> > > > > ### Author Response · Authors · 2024-08-14
> > > > >
> > > > > Dear Reviewer yDqR,
> > > > >
> > > > > Thank you very much for your consideration of our replies and for the rating increase to an acceptance score. We're very pleased to know that our response has addressed your concerns. Please kindly let us know if there are any further questions. Thanks again.

---

### Official Review · Reviewer_FpxG · 2024-07-12

**Soundness:** 4
**Presentation:** 3
**Contribution:** 3
**Rating:** 7
**Confidence:** 4

**Summary:**

The paper studies an under-explored graph anomaly detection problem where the detection models have access to a set of labeled normal nodes. To tackle this problem, it introduces a generative approach namely GGAD that generates pseudo anomaly nodes, called outlier nodes, to support the training of a discriminative one-class classifier. The key idea underlying this approach is to generate the outlier nodes in a way that can well simulate real anomaly nodes in both graph structure and feature representation perspectives. To achieve this, GGAD defines and incorporates two priors, including asymmetric local affinity and egocentric closeness, into its optimization objectives, with the former prior focusing on the alignment on the graph structure aspect and the latter on the feature representation aspect. The method is evaluated on six large real-world datasets and shows impressive detection performance compared to existing state-of-the-art methods.

**Strengths:**

- The paper is generally well-written and easy-to-follow.
- The problem setting is practical since labeled normal samples are easy to obtain in many real-world applications. Compared to the commonly studied unsupervised setting, this semi-supervised setting often results in better detection performance.
- The proposed method GGAD is novel. There have been many generative anomaly detection methods, but as far as I know, they are unable to consider the graph structure and the neighboring nodes’ representations. By introducing the two new priors, GGAD addresses this issue well. Fig.1 and Fig. 3 help demonstrate this effect.
- The method is compared with a range of unsupervised and semi-supervised methods on 6 real-world datasets with diverse genuine anomalies, and gains largely improved detection performance over these competing methods.
- The ablation study is plausible and justifies the contribution of each proposed prior.

**Weaknesses:**

- The outlier node generation in GGAD may cause non-trivial computational overhead.
- Despite better performance than the competing methods, GGAD gains an AUC of only around 0.6 on some datasets, such as DGraph and Reddit.
- In Fig. 4 (b), GGAD shows a fast AUPRC growth with increasing training size, but the other methods have a flat performance trend. What would be the reason behind?

**Questions:**

See the weakness

---

> ### Author Rebuttal · Authors · 2024-08-07
>
> Thank you very much for the constructive comments. We are grateful for the positive comments on our studied problem, technical contribution, and empirical justification. Please see our detailed response below
>
> > **Weaknesses #1**  The generation may cause non-trivial computational
>
> We agree that GGAD may cause computational overhead due to the outlier node generation. However, the overhead is small. This is justified by the time complexity analysis and running time statistical results in Appendix D, where GGAD runs much faster than many competing methods and it is comparable efficiency to the remaining methods.
>
> > **Weaknesses #2**  Low AUC on Reddit and DGraph
>
> Thank you very much for the comment. Reddit and DGraph are two challenging datasets in GAD. DGraph is a very large-scale graph that includes more than millions of nodes where the anomalies only account for 1.3 \%.  The fully supervised methods only yield an average of 0.02/0.67 in AUPRC and AUROC on this dataset [Ref1-2]. Similarly, Photo is a user-subreddit graph network on which fully supervised methods only yield an average of 0.06/0.65  in AUPRC and AUROC  [Ref1-2].
> Although our GGAD achieved low AUROC/AUPRC on these two datasets, not as high as the other datasets, it still shows good improvement compared to other state-of-the-art unsupervised and semi-supervised methods.
>
>
>
> > **Weaknesses #3**  The reason behind the flat performance of other methods
>
> The main reason is that these competing methods are trained based on their unsupervised proxy GAD tasks, so they have limited capability to increase their discriminability with increasing the number of normal nodes.
> On the contrary, our GGAD utilizes partially labeled normal nodes and two important anomaly priors to generate outlier nodes as negative samples to train a discriminative one-classifier. As the number of normal nodes increases, the generated outliers become more diverse, closely aligning with the real abnormal nodes in the dataset, thereby resulting in better discriminability and thus better AUPRC.
>
> **References**:
>
> - [Ref1] GADBench: Revisiting and Benchmarking Supervised Graph Anomaly Detection, NIPS2023
> - [Ref2] BOND: Benchmarking Unsupervised Outlier Node Detection on Static Attributed Graphs, NIPS2022

---

> ### Comment · Reviewer_FpxG · 2024-08-11
> **The reply has addressed my questions.**
>
> The reply has addressed my questions.

---

> > ### Author Response · Authors · 2024-08-12
> >
> > We're very pleased that our response has addressed your questions. Thank you very much for the positive comments and appreciation of our work.

---

### Author Rebuttal · Authors · 2024-08-07

Dear All Reviewers,

Thank you very much for the time and effort in reviewing our paper, and for the constructive and positive comments. Our rebuttal consists of two parts: **Global Response** where we address shared concerns from two or more reviewers and **Individual Response** where we provide a detailed one-by-one response to address your questions/concerns individually.

> ### Global Response to Shared Concern #1 More empirical evidence to verify the two priors and the second prior has not been empirically verified.


The asymmetric local node affinity prior,  ``the affinity between normal node is stronger than that between normal and abnormal nodes``, has been revealed in multiple recent studies on a range of datasets [Ref1-3]. To further verify this prior,  we provide more affinity visualization results on other GAD datasets including Amazon, Reddit, Elliptic, and Photo, as shown in Fig.1 in the uploaded **pdf**. The results show that the normal nodes have a much stronger affinity to its neighboring normal node than the abnormal nodes.

For the egocentric closeness prior,  ``the feature representations of outlier nodes should be closed to the normal nodes that share similar local structure as the outlier nodes``, we verify this prior by analyzing the similarity between normal and abnormal nodes based on the raw node attributes on the other four datasets in Fig.2 in the uploaded **pdf**. The results show that the real abnormal nodes can exhibit high similarity to the normal nodes in terms of local affinity in the **raw attribute space**. The main reason is that some abnormalities are weak or the existence of adversarial camouflage that disguises abnormal nodes to have similar attributes to the local community. This is the key intuition behind the second prior.


> ### Global Response to Shared Concern #2 The direction of two constraints seem to be contradictory to each other.

$L_{ala}$ and $L_{ec}$ are collaborative constraints rather then conflicting ones. $L_{ala}$ is designed to make the generated outliers have asymmetric affinity separability from normal nodes from the graph structure perspective, while $L_{ec}$ is devised to pull the representation of generated outliers closed to the node representations in its egonet in the feature representation space.  If we solely apply $L_{ala}$， it may generate some trivial outliers that are far from the normal node, see Fig.3(a) in the paper.  Although these trivial outliers share some local affinity property as the abnormal,  they have adverse effects on training a compact, discriminative one-class classifier in the feature representation space. Thus, we further introduce the egocentric closeness prior-based loss $L_{ec}$.  It enables the generated nodes to be close to the distribution of normal nodes. This joint force will result in outlier representations that are at the fringe of the normal node representations while structurally separable, preventing the generation of trivial outliers that are far away from the normal nodes.  This collaborative effect can also be observed in Fig.3 (a)-(c) in the paper. To further demonstrate the collaboration between these two prior-based losses,  we visualize the optimization of loss during the training in Fig.3 in the uploaded **pdf**, where 'ala' and 'ec' represent the two priors losses and the 'total' represents the sum of these two priors and BCE loss. From the results, we can see the two prior losses and the total loss are continuously decreasing and converging finally, further indicating that these optimizations are collaborative, resulting in an effective one-class discriminator.

Note due to the space limitations,  all the visualization figures of other datasets will be provided in the final paper.

As for **Individual Response**, we have provided a detailed one-by-one response to answer/address your questions/concerns after your individual review.

We very much hope our response has clarified the confusion, and addressed the concerns. We're more than happy to take any further questions if otherwise. Please kindly advise!

**Due to the space limitation, here we put a table for addressing Review JEU8 Question #2**
```
Table A1.  Comparison with the unsupervised GAD methods that use our GGAD-generated outliers (AUPRC/AUROC).
```
|**Data**        |**Amazon**          |**T-Finance**    |**Elliptic**    |
|:----           |:----:              |:----:           |:----:           |
| #Anomalies/#Top-K Nodes   |   387/500          |  351/1000       | 1448/2000       |
|  DOMINATE       |   0.1315/0.7025        | 0.0536/0.6087       | 0.0454/0.2960   |
|  GGAD-enabled DOMINATE|   **0.3462/0.8186**    | **0.0585/0.6275**   | **0.0613/0.2986**   |
|  OCGNN          |   0.1352/0.7165        | 0.0392/0.4732       | 0.0616/0.2581   |
|  GGAD-enabled OCGNN  |  **0.3950/0.8692**     | **0.0480/0.5931**   | **0.0607/0.2638**   |
|  AEGIS          |   0.1200/0.6059        | 0.0622/0.6496       | 0.0827/0.4553   |
|  GGAD-enabled AEGIS   |   **0.3833/0.8395**    | **0.0784/0.7024**   | **0.0910/0.5036**   |
|  GGAD           | 0.7769/0.9431          | 0.1734/0.8108       | 0.2484/0.7225   |


**References**:

- [Ref1] Addressing Heterophily in Graph Anomaly Detection: A Perspective of Graph Spectrum, WWW2023
- [Ref2] Truncated Affinity Maximization: One-class Homophily Modeling for Graph Anomaly Detection, NIPS2023
- [Ref3] Graph anomaly detection with bi-level optimization, WWW2024

---

### Decision · Program_Chairs · 2024-09-25

**Decision:**

Accept (poster)

**Comment:**

This paper proposes a semi-supervised graph anomaly detection method that enhances accuracy by effectively generating outlier nodes. The reviewers appreciated the well-motivated problem setting and recognized the high originality of the proposed approach, as well as the comprehensive experiments conducted. Given the positive reviews from all reviewers, I recommend acceptance.